# Insights into the North Hemisphere daily snowpack at high resolution from the new Crocus-ERA5 product

Silvana Ramos Buarque [1], Bertrand Decharme [1], Alina L. Barbu [1], and Laurent Franchisteguy [2]

[1]Météo-France, CNRS, Univ. Toulouse, CNRM, Toulouse, France
[2]Météo-France, Direction des Systèmes d'Observation, Toulouse, France

**Correspondence:** S. Ramos Buarque (silvana.buarque@meteo.fr)

**Abstract.** This article provides an overview of the daily Crocus-ERA5 snow product covering the Northern Hemisphere from 1950 to 2022. It assesses the product's performance in terms of snow depth and cover compared to in situ observations and satellite data. Compared to its predecessor, Crocus-ERA-Interim, Crocus-ERA5 benefits from improved spatial resolution and better atmospheric data assimilation, resulting in more accurate snowpack estimates, especially during spring in Eurasia. The findings show a good match with observations, though biases remain, particularly in some Arctic regions, where the model tends to overestimate spring melt. In low-vegetation areas such as tundra, Crocus-ERA5 may introduce biases due to its limited consideration of interannual vegetation changes, leading to inaccuracies in the simulation of snowmelt. The production of this snow dataset responds to the request of the continental cryosphere community. In particular the French and Canadian government institutions CNRM (National Center for Meteorological Research) and ECCC (Environment and Climate Change Canada) have been involved in monitoring Arctic snow cover as part of the "Terrestrial Snow" section of the Arctic Report Card since 2017. The Crocus-ERA5 product is freely available on a daily basis and at 0.25° resolution over the 1950-07-01 to 2023-06-30 period (Decharme et al., 2024, https://doi.org/10.5281/zenodo.14513248).

## 1 Introduction

The Arctic is particularly vulnerable to global climate change (Druckenmiller et al., 2024). Changes in atmospheric circulation can result in altered precipitation patterns affecting the amount and type of precipitation that contributes to snow accumulation (Cai et al., 2024; Ramos Buarque and Salas y Melia, 2018; Vihma et al., 2016). Whereas changes in sea ice can also indirectly affect the land surface snow cover by altering the surface albedo and heat exchange processes (Pörtner et al., 2019; Letterly et al., 2018). Over the past few decades, the Arctic has experienced warming at a rate approximately twice as fast as the global average, resulting in unquestionable alterations to the Arctic cryosphere, as documented in the Intergovernmental Panel on Climate Change (IPCC) Special Report on Oceans and Cryosphere in a Changing Climate (Meredith et al., 2022). In addition to the above-average surface temperatures, an unprecedented (since instrumental records began) geographic spread of heat waves and warm spells occurred (Dunn et al., 2023). As a result, the rise in global temperatures linked to oceanic warming has led to a reduction in sea ice extent (Landrum and Holland, 2022).

The pronounced decline in sea ice extent during late summer and early autumn increases the ocean's heat storage capacity, which in turn delays the formation of new ice at the onset of the cold season. This additional ocean heat also influences adjacent land areas, contributing to unprecedented high near-surface permafrost temperature (Screen and Simmonds, 2010). As a result, the active layer (the seasonally thawed layer of soil above the permafrost) is becoming deeper, with serious implications for soil stability, surface hydrology, and carbon emissions, all of which further intensify climate warming(Wang et al., 2021; Luo et al., 2016; Biskaborn et al., 2019; Natali et al., 2019). On land, terrestrial snow cover, which now vanishes entirely from Arctic land surfaces during summer, plays a critical role in this interconnected system. Snow acts as both an insulator for the underlying permafrost (Decharme et al., 2016; Matthes et al., 2025) and a reflective surface that regulates land surface temperatures. Its disappearance exposes the ground to direct solar radiation, accelerating permafrost thaw and deepening active layers (Zhang, 2005; Pulliainen et al., 2020). In late spring, early snow loss reduces surface albedo, allowing more solar energy to be absorbed and amplifying surface warming (Flanner et al., 2011; Brown and Mote, 2009). This accelerates snowmelt and reinforces the snow-albedo feedback, a key driver of permafrost thaw, shifts in runoff patterns, changes in wildlife behavior, and heightened wildfire risk (Callaghan et al., 2011; Biskaborn et al., 2019). Snow loss during this period also enhances carbon uptake in subsequent months, affecting Arctic ecosystem productivity by increasing vegetation productivity and extending the growing season, thereby affecting Arctic ecosystem dynamics (Romanovsky et al., 2010; See et al., 2024). Moreover, snow responds rapidly to temperature fluctuations, particularly in early autumn and early spring, making it a highly sensitive and visible indicator of Arctic climate change. Variations in snow cover not only influence the timing and volume of snow melt into rivers and streams but also affect permafrost stability and the functioning of Arctic ecosystems (Zhang, 2005). Together, these interlinked changes in sea ice, snow, and permafrost create reinforcing feedback loops that amplify the pace and impacts of climate change in the Arctic (Screen and Simmonds, 2010; Druckenmiller et al., 2024).

The aforementioned evidences clearly illustrates the intricate nature of these changes and the complex interactions between the various components of the Arctic climate system. In this context, the Arctic Report Card (ARC, https://arctic.noaa.gov/report-card/), published annually by the National Oceanic and Atmospheric Administration (NOAA) of the United States, enables the global scientific community to monitor and document environmental changes in the Arctic (NOAA Arctic Program, ongoing; Druckenmiller et al., 2024). This report presents a comprehensive and up-to-date assessment of the current state of the Arctic, based on the most recent scientific data ans coordinated expert analysis. The ARC addresses a broad range of environmental indicators, including air and ocean temperatures, precipitation, Greenland ice sheet mass balance, sea ice extent, snow cover, permafrost conditions, and vegetation trends. The objective of the report is to provide policymakers, scientists, and the general public with information about the rapid and widespread changes occurring in the Arctic as a result of climate warming (Overland et al., 2019; Druckenmiller et al., 2024). It identifies significant events of the past year, interannual variability, and long-term trends, providing robust and accessible monitoring of the evolution of this critical region for the global climate. By providing accurate and timely information, the ARC plays an important role in raising awareness and supporting informed decision-making on the environmental challenges facing the Arctic.

Since 2017, the French National Center for Meteorological Research (CNRM) has been involved in the "Terrestrial Snow" contribution to the ARC through a collaboration led by the Environnement et Changement Climatique Canada (ECCC) Institute

to monitor the evolution of snow cover and mass each year (e.g. Mudryk et al., 2024). CNRM's role is to provide a daily snowpack product derived from the Crocus complex snow scheme (Brun et al., 1992, 1989; Vionnet et al., 2012) contributing to the monitoring and advancement of knowledge on the snow cover and mass of the North Hemisphere (NH). In addition to its use in the ARC, from 2017 to 2020, this product has also been used in several scientific studies (Mudryk et al., 2015; Mortimer et al., 2020; Kouki et al., 2023).

Recently, our snow product has been updated using the same Crocus configuration but driven by the fifth generation of the ECMWF global atmospheric reanalysis ERA5 (Hersbach et al., 2020) , referred to as Crocus-ERA5. Although this product has been little evaluated, it has been used since 2021 in the ARC report (Mudryk et al., 2024) and in several scientific studies (Derksen and Mudryk, 2023; Mudryk et al., 2025).

The aim of this study is to provide a brief evaluation and insight into this Crocus-ERA5 daily snow product, which support numerous studies on the evolution of the Arctic snow cover in the coming years. This product is the result of a novel rerun of the Crocus snowpack model whose innovation is threefold: (i) the use of the improved ECMWF reanalysis ERA5 as atmospheric forcing, offering higher spatial and temporal resolution, which enhances the accuracy of snowpack simulations; (ii) comprehensive documentation of this update, highlighting both the strengths and limitations of the two most widely used variables in the community across the vast expanse of the Northern Hemisphere; and (iii) validation through comparison with fully independent references, including in-situ observations and multi-source data analysis.

The data and methods used to evaluate the new Crocus-ERA5 product are presented in Section 2. The main findings of this study are presented in Section 3. The Crocus-ERA5 product is first compared with its predecessor, Crocus forced by ERA-Interim, hereafter referred to as ERAI, to assess the progress (or lack thereof) between successive generations of these snow products. In the Arctic region, validation of the snowpack benefits of various observations and estimates, including in situ and satellite data, so Crocus-ERA is then evaluated against a range of in situ and satellite data. Finally, a brief discussion and the main conclusions are presented in section 4.

## 2 Data and Methods

The performance of snowpack modelling in the context of climate change, can be effectively characterized using two key variables : snow depth and snow cover. These variables are widely used as climate change indicators because of their strong interactions and feedbacks with the surface energy balance.

Snow depth plays a critical role in both seasonal and long-term evolution of frozen ground and permafrost (Zhang, 2005; Park et al., 2016). Changes in snow depth have significant implications in atmospheric circulation and  may influence weather in other parts of the world (Cohen et al., 2014). For instance, large snow depths in western Russia have been associated with local cyclonic circulation, where northerly flow to the west of the trough favour the advection of cold air into the regions, helping to preserve the snow cover (Peings and Magnusdottir, 2015). Locally, over regions at higher latitudes or altitudes the dynamics of snow depth may favour more prolonged snow cover periods , further modulated by local environmental conditions (Brown and Mote, 2009). Extreme weather events such as blizzards, can alsolead to rapid changes in snow depth over short

time scales (Déry and Yau, 2001). Due to its climatic importance, considerable effort has been made over the years by various countries in the Northern Hemisphere to carry out extensive and consistent measurements of snow depth (Zhong et al., 2018).

Snow cover is an equally important climate change proxy in the Northern Hemisphere. Its seasonal cycle drives a number of important energy and water cycle processes (Brown and Mote, 2009). In Arctic regions, snow cover dynamics drive the seasonal thermal regime of the ground, with implications for the carbon cycle, permafrost, and terrestrial and freshwater ecosystems (Zhong et al., 2018; Park et al., 2016). In particular, during spring, the retrat of snow cover directly influence the magnitude of climatic warming (Flanner et al., 2011; Qu and Hall, 2007).

Another key snow-related variable widely used by the scientific community is snow water equivalent (SWE), which represents the amount of water contained in the snowpack (Pulliainen et al., 2020; Mudryk et al., 2020, 2025, 2024) . SWE is particularly valuable because it provides a direct measure of the snowpack's total water content, which is crucial for hydrological applications such as water resource management and runoff prediction (Sturm et al., 2010; Dozier et al., 2016) . In the Crocus-ERA5 model, snow depth accounts for the density of individual snow layers, while SWE is derived from both snow depth and density (Vionnet et al., 2012) . Snow depth is relatively easy to observe through remote sensing and ground measurements, making it a valuable proxy for validating snowpack models. In contrast, SWE is more difficult to measure directly. Satellite-derived snow depth often relies on SWE estimates combined with assumed or modelled snow density, an approach that introduces significant uncertainties due to spatial variability in density, retrieval errors, and sensor limitations. These factors can lead to biases in satellite-based snow depth products (Derksen et al., 2005; Larue et al., 2017; Pulliainen et al., 2020) . However, in-situ measurements provide consistent and traceable data. They are widely used to evaluate snow models such as Crocus-ERA5. Snow depth is easier to measure than SWE and is available from many long-term monitoring programs in the Northern Hemisphere (Brown and Mote, 2009; Zhong et al., 2018). These records cover several decades and large geographic areas. They are essential for detecting trends and variability. Snow depth observations are also used to improve SWE estimates and to support hydrological and climate modelling (Mudryk et al., 2015; Mortimer et al., 2020). Snow cover extent complements snow depth. It describes the spatial distribution of snow, which influences the surface energy balance and the timing of melt (Flanner et al., 2011; Qu and Hall, 2007). In this study, both snow depth and snow cover are compared with independent observations to assess the reliability of Crocus-ERA5.

## 2.1 Atmospheric Forcings

The Crocus snowpack model is forced, in this study, to simulate snow evolution with the ERA5 atmospheric reanalysis (Hersbach et al., 2020), produced by the European Centre for Medium-Range Weather Forecasts (ECMWF). As the latest global meteorological reanalysis, ERA5 represents a significant advancement over ERAI (Dee et al., 2011). Table 1 summarizes the main differences between ERAI and ERA5 reanalyses. One of the most notable improvements is the spatial resolution, with ERA5 offering a resolution of approximately 31 km, compared to 80 km in ERAI. Beyond the finer spatial and temporal resolution, ERA5 also benefits from over a decade of advancements in model physics, core dynamics, and data assimilation.

The ERA5 reanalysis includes an advanced land data assimilation system that has been enhanced to improve the representation of hydrological variables such as soil moisture and snow depth. Both ERAI and ERA5 use a single-layer snow scheme

that assumes uniform density and temperature throughout the entire snowpack. Compared to ERAI, ERA5 improves the representation of key physical processes, including snow accumulation, melting, wind-driven redistribution as interactions with the ground by incorporating better data assimilation techniques. However, ERA5 does not account for vertical variations within the snowpack, such as temperature gradients or changes in snow properties at different depths - features typically represented

130 in multi-layer snow models. Consequently, snow parameters derived from ERA5, including snow depth and SWE, are less detailed and refined than those produced by multi-layer models that explicitly resolve multiple snowpack layers to capture more detailed variations in snow properties.

**Table 1.** Key differences between ERA-Interim (ERAI) and ERA5 reanalysis products.

| Feature | ERAI | ERA5 |
|---|---|---|
| Reference | Dee et al. (2011) | Hersbach et al. (2020) |
| Time coverage | 1979–2019 | 1950–present |
| Horizontal resolution | 0.75° ( 80 km) | 0.25° ( 31 km) |
| Temporal resolution | 6-hourly data | Hourly data |
| Vertical levels | 60 levels (top at 0.1 hPa) | 137 levels (top at 0.01 hPa) |
| Data assimilation system | 4D-Var (IFS Cycle 31r2) | 4D-Var (IFS Cycle 41r2) |
| Land surface model | TESSEL | HTESSEL (updated version) |
| Snowpack structure | Single-layer scheme | Single-layer scheme |
| Advantages | Longer-established dataset, widely validated in past studies | Higher resolution, more detailed atmospheric processes, updated physics, improved land-surface interactions |

## 2.2 Crocus-ERA5 Framework

Crocus is coupled to the ISBA (Interactions between Soil–Biosphere–Atmosphere) land surface model (Vionnet et al., 2012;

135 Brun et al., 2013) and embedded in the SURFEX numerical platform (Masson et al., 2013, https://www.umr-cnrm.fr/surfex/). This coupling, as described by Vionnet et al. (2012), allows Crocus to interact with other surface components such as soil, low vegetation, and atmosphere within a consistent framework. Through the ISBA model, Crocus is forced by key atmospheric variables, including 2m air temperature and humidity, liquid and solid precipitation, incoming shortwave and longwave radiation, and wind speed. Integrated into SURFEX, Crocus benefits from a shared surface energy balance and atmospheric forcing

140 structure, enabling more realistic simulations of snowpack evolution in complex surface environments. A full description of the Crocus model and the technical configuration of the Crocus Northern Hemisphere Snowpack product can be found in Vionnet et al. (2012) and Brun et al. (2013), respectively.

The demand for improved snow modelling increased in the early 2000s, driven by growing interest in climate change impacts and water resources (Brown and Mote, 2009; Callaghan et al., 2011; Flanner et al., 2011). In the early 2010s, the availability

of high-resolution atmospheric reanalysis datasets encouraged the application of one-dimensional multi-layer mass and energy balance models over large are (Vionnet et al., 2012; Brun et al., 2013). These datasets improved the representation of snow season dynamics and refined albedo estimates through explicit modelling of snow grain types. Enhanced atmospheric forcing from reanalyses such as ERA5 has further increased model accuracy, particularly in representing snow processes and land-atmosphere interactions (Hersbach et al., 2020).

Until 2020, the Crocus model was driven by meteorological forcing (temperature, precipitation, humidity, wind, etc.) derived from the ERAI global atmospheric reanalysis. Building on the availability of more advanced reanalysis products, the Crocus-ERA5 product was developed. A set of seven snow-related variables (snow depth, SWE, liquid water content in the snowpack, snow albedo, snow surface temperature, snowpack internal temperature, and snow cover fraction) is freely available on a daily basis at https://doi.org/10.5281/zenodo.14513248 (Decharme et al., 2024).

The Crocus-ERA5 dataset also benefits from adopting the standards of the Coupled Model Intercomparison Project Phase 6 (CMIP6), including metadata structured according to ISO 19115, CF metadata conventions, and compliance with the Data Reference Syntax (DRS), practices that promote interoperability and robust data management (Eyring et al., 2016). Additionally, CMIP6 data follows the Climate and Forecast (CF) metadata conventions (CF-1.7, CMIP-6.2) and complies with the Data Reference Syntax (DRS) for systematic file organization and naming, promoting interoperability (Juckes et al., 2020).

## 2.3 Snowpack modelling

The Crocus multi-layer snow model simulates snow albedo, heat transfer and phase change, snow mass, snow density, and snow grain metamorphism based on experimental laws (Brun et al., 1989, 1992; Vionnet et al., 2012). The number of layers is variable from 1 to 50 depending on snow depth and stratification.

The snowpack model does not explicitly represent forested areas. Instead, these zones are treated as low vegetation, following the approach of Brun et al. (2013). This simplification was a deliberate choice, as the model does not simulate snow-forest interactions. While this improves internal consistency, it comes with some limitations. Forests significantly impact snow processes through canopy interception, sublimation, and shading effects (Napoly et al., 2020; Monteiro et al., 2024; Essery et al., 2025). Ignoring these mechanisms can lead to overestimation of snow accumulation at ground level, as it neglects snowfall interception by tree canopies and underestimates sublimation of intercepted snow driven by longwave (infrared) radiation. Forest shading also reduces surface albedo, slowing snowmelt. Because these interactions are complex and remain challenging to simulate accurately (Napoly et al., 2020; Essery et al., 2025), Crocus-ERA5 adopts a simplified representation of the land surface, simulating the snowpack over an idealised grassland cover with climatological physiography (e.g., no interannual variability in leaf area index).

Rising temperatures and increasingly frequent winter thaws are altering snowpack structure and distribution, making it essential for models to represent these changes through explicit physical processes (Flanner et al., 2011; Brown and Mote, 2009). Crocus-ERA5 addresses this by simulating snowpack evolution in detail, including the snow cover fraction (SCF), and by distinguishing snow-covered from snow-free areas (e.g., vegetation or bare soil) within each grid cell (Brun et al., 2013). This separation limits unrealistic vegetation–snow interactions and improves the simulation of snowmelt and rain infiltration.

To further enhance realism, the model dynamically represents the vertical structure of the snowpack, dividing it into between 3 and 50 layers depending on depth and stratification (Vionnet et al., 2012). This layered approach enables the capture of thermal gradients and the conservation of mass and heat during melting or accumulation, while representing key processes such as solar absorption, water retention, heat transfer, and compaction. In thin snowpacks, solar radiation can penetrate to the ground, affecting albedo, whereas in deep snowpacks it is absorbed within the snow, which then acts as an insulator that reduces heat conduction, limits turbulent exchange, and alters the net radiation budget due to its high albedo. Consistent with these physical processes, two distinct regimes of SCF behavior coexist, showing that both shallow and deep snow can reach full coverage ($SCF \approx 1$), but through different mechanisms. Scatterplots of Crocus-ERA5 snow depth versus SCF for Eurasia and North America (not shown) reveal this dual-regime relationship clearly. At shallow snow depths ($< 0.2m$), SCF spans the full range ($0 to 100\%$) but changes only slowly with depth (low slope). This reflects the patchy and uneven nature of shallow snow cover: adding a few centimeters of snow does not always increase coverage. At greater depths ($> 0.2m$), SCF rises much more rapidly toward full coverage (steeper slope). This indicates the formation of a continuous snowpack, where additional snow quickly fills in bare areas and leads to nearly complete coverage. In this way, Crocus-ERA5 realistically captures the transition between patchy residual snow and a continuous snowpack.

To represent the impact of snow on the surface radiation budget, the model diagnoses an effective land surface albedo from the difference between the downward radiative fluxes and the net radiative flux at the surface. In snow-covered areas, and particularly when snow overlays vegetation, upwelling shortwave and longwave radiation can markedly reduce the effective albedo (Boone et al., 2017).

## 2.4 Analysis Methods

Evaluate changes in the estimation of the snowpack from Crocus stand alone involves variations influenced by climate change that are directly taken into account by atmospheric forcing. The two variables retained in our study, snow depth and snow cover, are strongly linked to each other and reflect the direct response of atmospheric forcing providing essential information on the performance of the snowpack modelling. For this reason, the new Crocus-ERA5 product with a horizontal resolution of 1/4° is first compared with the previous Crocus-ERAI version with a resolution of 1/2° over the period 1979-2018. The comparison focuses on the monthly variability of snow depth anomalies. It evaluates changes in Arctic snow cover variability due to changes in atmospheric forcing, and to a lesser extent, spatial resolution, separately for North America and Eurasia, as Eurasia, North America, and Greenland are affected differently by the components of the cryosphere. Within the Arctic Circle, the Eurasia pan-region includes Scandinavia (Norway, Sweden, Finland) and northern Russia, while North America includes northern Alaska (USA) and northern Canada.

In this study, the assessment of the Crocus-ERA5 daily snow depth is done against the harmonized dataset of in-situ observations across North America and Eurasia providing a comprehensive view of historical snow conditions. In Brun et al. (2013), this *Crocus-ERAI* daily snow product was validated against local observations from over 1000 monitoring stations in northern Eurasia. Assessing the timing of snow onset and melt simulated by large-scale models is especially challenging due to the rapid and highly variable response of snow to temperature fluctuations. The ability of the Crocus-ERA5 to map snow depth

(or mass, related to depth by the density) is thus evaluated against these in-situ observations in terms of time-average and local daily variability. To identify particular errors in Crocus-ERA5, we analyzed diagnostics derived from continuous snow depth information, including the duration of snow and the first and last day of continuous snow, following the method described in many previous studies (Brun et al., 2013; Schellekens et al., 2017; Decharme et al., 2016, 2019). For each diagnostic, we calculated the correlation coefficient, bias, and centered root mean square error between model and observations, along with the number of available paired values. Additional diagnostics include the average and annual maximum snow depths, the date of maximum snow depth (expressed as days since January 1st) as the number of snow-covered days per year.

Over the years, significant advancements in snow cover analysis, particularly through multi-sensor approaches, have greatly improved mapping by reducing the uncertainties associated with sparse, single-sensor data. Issues related to mismatches in temporal and spatial scales - which historically made large-scale validation increasingly reliant on modelling frameworks and gridded interpolations - have thus been reduced.

The use of snow cover extent (SCE) provides a distinct advantage, as it represents a cumulated variable that can be reliably derived from satellite observations over large regions. Additionally, expressing snow cover as a percentage (%) offers a valuable advantage for validation purposes, providing a simple, spatially explicit measure of snow presence or absence over a given area. SCE and snow cover percentage can thus be directly compared with satellite-derived estimates, making them practical and robust metrics for evaluating the spatial accuracy of snowpack models.

In this study, SCE is characterized by representing its monthly standardized anomalies for 2000-2022, which directly addresses the main objective of validating the seasonality and interannual variability of snow cover changes over large areas.

## 2.5 Observational Data

The in-situ dataset includes over 2,000 stations with daily observations spanning the period from 1950 to 2012. Most measurements are taken at synoptic stations following World Meteorological Organization (WMO) standards, typically representing bare ground or open areas with regular grass cutting (Schellekens et al., 2017, https://doi.org/10.5194/essd-9-389-2017-supplement). This extensive dataset that was carefully processed to ensure consistency and minimize elevation-related biases offers valuable spatial and temporal coverage, providing robust insights into historical snow depth variability and trends. Decharme et al. (2019) described the processing to this dataset namely outlined specific selection criteria, including: (i) a local-model elevation of less than 100 m, (ii) a minimum of 100 days with a non-zero depth measurement over the full period and, (iii) at least eight snowy days per year. While such site selection ensures good consistency with the open-field grassland configuration of Crocus-ERA5, it does not guarantee full spatial representativeness of a 0.25°grid cell. Sub-grid heterogeneities such as partial forest cover, slope, aspect, or wind-driven redistribution may still locally influence the comparison between observations and simulations.

The ability of the Crocus-ERA5 to map snow cover is evaluated against the Interactive Multisensor Snow and Ice Mapping System (IMS) satellite data (U.S. National Ice Center, 2008, https://nsidc.org/data/g02156/versions/1). The IMS snow cover analysis incorporates estimates from NOAA AVHRR, MODIS, VIIRS, and Sentinel satellites, alongside in-situ observations and NCEP model data (Helfrich et al., 2007). By integrating time-sequenced satellite imagery, IMS enhances the differentiation

between snow and cloud (Estilow et al., 2015), ensuring superior spatial and temporal coverage for more precise and reliable snow cover estimate. The adoption of advanced multi-sensor fusion techniques in 2020 has set IMS apart, allowing it to combine multiple satellite sources and models for an all-weather, day-and-night analysis. This makes IMS more robust and comprehensive than other snow cover analyses, which often lack the same level of real-time updates and multi-source data integration. Additionally, in this study, the period covered by IMS complements that of in-situ observations (1950-2012). To align as closely as possible with model resolution, IMS snow cover data is used at a spatial resolution of 24 km for the period 2000-2022.

However, IMS is not free from uncertainties. Potential errors can arise from cloud-snow discrimination challenges, coarse spatial resolution at high latitudes, and differences in input data availability between regions and seasons, occasionally leading to misclassification of snow-covered areas, especially during transitional periods (Helfrich et al., 2007; Estilow et al., 2015). Nevertheless, IMS remains one of the most consistent and widely used multi-sensor products for large-scale snow cover monitoring, making it suitable for the objectives of this study.

## 3    Crocus-ERA5 assessment

While Crocus-ERA5 provides valuable insights into snowpack dynamics, it faces limitations in accurately representing different land cover types. Forests are not explicitly represented and are treated as open grassland surfaces, which can lead to biases in mean snow depth due to the absence of canopy interception, sublimation, and shading effects (Napoly et al., 2020; Essery et al., 2025).However, as shown by the multi-product evaluation of (Mudryk et al., 2025), interannual variability in Crocus-ERA5 remains consistent with other high-skill products, reflecting the dominant influence of large-scale meteorological forcing on year-to-year snowpack changes. Beyond forests, Crocus-ERA5 faces limitations in open areas with low vegetation, such as grasslands and tundra, where it may not accurately capture wind-driven snow redistribution. This can introduce biases in snow depth and duration estimates, affecting the dataset's applicability in regions where blowing snow and drift processes play a dominant role in snowpack evolution.

Despite these challenges, Crocus-ERA5 enhances snow cover simulation by explicitly modelling snowpack processes and distinguishing snow-covered and snow-free areas, surpassing the capabilities of satellite observations alone. While satellites rely on indirect measurements and assumptions, Crocus-ERA5 uses physically based simulations to represent snowpack dynamics more accurately. Its detailed physical representation, combined with increasing spatial and temporal resolution, improves snowpack dynamics simulation, enhances accuracy, and extends dataset applicability across diverse landscapes.

### 3.1    Comparison with the Crocus-ERAI product

The variability in monthly snow depth differences between the two reanalyses is highlighted through the differences in their respective monthly anomalies, shown both annually and seasonally for the Eurasian and North American regions (Figure 1). The monthly anomalies are computed by subtracting the respective monthly climatology (1979-2018) from each dataset; that is, each month is compared to its own long-term monthly mean (e.g., May 2016 is compared to the May climatology).

The anomaly differences are then obtained by subtracting one set of monthly anomalies from the other, allowing for a direct comparison of interannual variability between the two reanalyses.

Unlike the anomalies from the individual datasets (which typically range between -0.1 m and +0.1 m, not shown), the anomaly differences span a narrower range (-0.03 m to +0.03 m). The resulting values represent how much the two reanalyses diverge in their depiction of snow depth variability. Note that in this representation (anomaly of anomaly), values within the range of -0.01 m to +0.01 m are generally not significant, as they fall within the model's residual variability. This is particularly evident during the summer months, when snow depth is minimal and the remaining snowpack is limited to isolated areas where melting is slower. In this period, anomaly differences are primarily driven by the model's internal physics (such as energy balance, snow compaction, and metamorphism) rather than external climatic forcing. Outside of summer, larger anomaly differences are more likely to reflect meaningful discrepancies in the representation of snow accumulation and melt processes, and are therefore more indicative of how the models interact with external drivers like precipitation and temperature.

In Eurasia, the long-term decrease in snow depth observed since the early 21st century is reflected in more pronounced negative anomaly differences (rust shades), particularly in April and May of the years 2010, 2013, and 2016. These negative values indicate that ERA5 shows stronger snow loss than ERA-Interim during the spring melt period in those years, highlighting potential differences in how each reanalysis represents snow melt timing, energy balance, or snow accumulation earlier in the season. The persistent negative values in the difference between Crocus-ERA5 and Crocus-ERAI snow depth anomalies (Crocus-ERA5 – Crocus-ERAI) from 1999 to 2016 suggest a shift in the atmospheric forcing characteristics between ERA5 and ERA-Interim, particularly during the spring melt season. ERA5, with its higher spatial and temporal resolution and more extensive assimilation of satellite observations provides warmer and more radiatively active near-surface conditions compared to ERA-Interim, specially after 1999. When used as input to the Crocus snow model, these conditions can lead to an earlier or more rapid onset of snowmelt, resulting in systematically lower snow depth anomalies in Crocus-ERA5 relative to Crocus-ERAI. This behavior is consistent with the findings of (Brun et al., 2013), who demonstrated that Crocus, due to its detailed representation of snow stratigraphy and melt processes, tends to delay melt relative to reanalyses with simpler snow schemes. However, even within Crocus, differences in the atmospheric forcing, particularly in spring, can significantly influence the timing and magnitude of snowmelt, amplifying the divergence between simulations based on ERA5 and ERA-Interim. In North America, however, the similarity between the snow depth anomalies of both simulations is more nuanced, and there is a significant interannual variability throughout the full data series.

The snow depth on the ground is closely related with the extent of snow cover, particularly in terms of its spatial distribution. Furthermore, the anomaly of snow cover varies considerably depending on the geographic location and local climate conditions. In the late spring (April to June), rising temperatures cause a decrease in snow thickness and lead to snow melt. Conversely, in autumun (September to November), the return of colder conditions favours new snow accumulation, marking the onset of the seasonal increase in snow cover. Figure 2 presents the Arctic SCE anomaly index, calculated as standardized anomalies in SCE over land areas within the Arctic Circle (latitudes > 60° N) from 1979 to 2018, separately for the North American and Eurasian Arctic regions. The solid black and dashed lines depict 5-yr running means, illustrating smoothed trends, while the circles show yearly standardized anomalies relative to the 2018 baseline. Across both regions, the Arctic SCE anomaly

index to Crocus-ERA5 and the Crocus-ERAI exhibit a closely similar interannual variability, with an overall declining trend throughout late spring and early autumn months. In Eurasia, around 2012, a particularly pronounced reduction in snow cover occurred during late spring and early summer (April-June), preceded by another noticeable decline in the autumn months (September-November) around 2009. These patterns highlight the ongoing and broad seasonal impacts of Arctic warming on snow cover dynamics. Furthermore, since the 2000s, Crocus–ERA5 anomalies in the North American sector appear slightly less pronounced than those from Crocus–ERAI during May and June, suggesting a reduced sensitivity to late spring conditions in the newer reanalysis. In autumn, the differences in interannual variability between the two datasets are muted. The yearly standardized anomalies relative to 2018 (represented by circles) show a broadly consistent pattern across datasets, with coherent interannual variability. These anomalies reflect how the monthly snow cover state in each year compares to 2018 and deviates from the smoothed long-term climate trend. Notably, the magnitude of these deviations tends to be greater during the onset of snow accumulation, particularly in the Eurasian sector (Figure 2, panels g-l), highlighting a stronger influence of short-term atmospheric variability in early winter. In contrast, snow cover during late spring is more strongly shaped by long-term climate trends, which control the timing and rate of snowmelt. The Arctic SCE anomaly index effectively highlights consistent large-scale snow cover patterns by minimizing local biases, which explains why differences between the ERA5- and ERAI-driven Crocus simulations appear limited. While this diagnostic is robust for analyzing trends and interannual variability, it may not fully capture finer-scale or distributional differences. Complementary diagnostics, such as those based on full distributions, can provide additional insights into the influence of reanalysis forcing.

A more detailed comparison using the joint distribution of mean and standard deviation of snow depth in April and May over the 1979-2018 period (Figure 3), reveals notable differences between the two forcings. There is a clear stability in snow depth distribution, i.e. deviation varies similarly around the mean, for both regions Eurasia and North America. In Eurasia, the Crocus–ERA5 simulation shows a systematic shift toward both higher mean snow depth and greater variability indicating that ERA5 produces a snowpack that is not only deeper on average but also more heterogeneous. This pattern reflects a proportional relationship, where increased snow depth coincides with enhanced interannual variability.

In April, the mean monthly snow depth increases by about 15% in Eurasia (from 0.44 to 0.52 m) and 8% in North America (from 0.42 to 0.46 m) when using ERA5 instead of ERAI. The standard deviation rises in similar proportions: 15% in Eurasia (from 0.24 to 0.29 m) and 7% in North America (from 0.24 to 0.26 m). These parallel increases suggest a consistent upward shift in both the average and the spread of snow depth values with ERA5 forcing. In May, this shift becomes more pronounced. Mean snow depth is 26% higher in Eurasia (from 0.17 to 0.23 m) and 16% higher in North America (from 0.21 to 0.25 m). The standard deviation increases accordingly, by 24% in Eurasia (from 0.21 to 0.28 m) and 10% in North America (from 0.24 to 0.26 m). In North America, the smaller relative increase in variability suggests that snow depth values become more consistently clustered around higher means, with less variability in lower values. These proportional increases in both mean and standard deviation in Crocus–ERA5 suggest a shift toward more abundant and variable snowpacks. This is consistent with enhanced snow accumulation and reduced melt rates, likely driven by higher cold-season precipitation and lower spring temperatures in ERA5, particularly in mountainous and mid- to high-latitude regions (Wang et al., 2019). The comparable

spread and clustering of snow depth values suggest that wind-driven snow redistribution, a key factor influencing snow depth, is similarly represented in both reanalyses.

This behavior likely reflects the fact that, under snow-rich conditions, atmospheric fluctuations have a stronger influence on snow accumulation, resulting in greater variability. The higher resolution and improved atmospheric inputs in ERA5 likely enhance the representation of these variations, particularly in Eurasia, where the differences between simulations are more pronounced. These findings illustrate the added realism introduced by ERA5 forcing. In contrast, North American distributions remain more consistent across reanalyses, suggesting lower sensitivity to the choice of atmospheric forcing in that region.

## 355   3.2   Snow depth

The annual cycle of snow depth is characterized by a gradual accumulation of continental snow from October to March, followed by a rapid ablation of the snowpack during the spring (Figure 1). The snow that persists at the advent of spring incorporates precipitation that occurred during the preceding winter, thereby reflecting the atmospheric variability of winter conditions. To represent the onset of the spring period, the amount of snow is verified by calculating the time-averaged snow

depth on 10 March for two climatological periods for both the Crocus-ERA5 and the in-situ observations products. Two tier of climatic normals have been highlighted: the period 1950-1980 is used to define the past climate and the period 1980-2012 is used to define the present-day climate (Figure 4). The present-day period cover the start of the acceleration in global warming and particularly the rapid warming of the Arctic that has led to the thawing permafrost.

    Agreement between modelled (shading) and observed (circles) snow depths is indicated by matching colours (Figure 4,

left). On 10 March, Crocus-ERA5 generally reproduces the spatial distribution of snow depth across the Northern Hemisphere, but with a clear tendency to overestimate. The corresponding bias map for the same day (Figure 4, right) illustrates the differences between modelled and observed snow depth values. The legend of this map was carefully designed to reflect the actual distribution of these biases. To highlight meaningful discrepancies while reducing the influence of near-zero noise, we grouped bias values into four representative classes: (0.05-0.1), (0.1-0.3), (0.3-0.8), and > 0.8 m. This binning scheme captures

the core of the distribution, with over 60 % of the data falling below 0.1 m. All biases in this range are positive, indicating a consistent overestimation of snow depth by the model. This classification improves visual interpretation by clearly identifying areas of stronger overestimation, particularly where snow is modelled but not observed. Small localized bias (red tones) are mainly associated with snow-free areas-such as the western coast of North America, the Rockies, and parts of eastern Asia-where the model simulates snow that is not present in observations. In contrast, greater than 0.1 m, shown in light tones, indicate

good agreement between modelled and observed values, especially across central Eurasia and some coastal zones. Overall, the spatial distribution of biases on 10 March highlights the ability of Crocus-ERA5 to capture large-scale snow patterns while also revealing systematic overestimations and regional limitations, especially in areas with complex topography.

    Note that there is good agreement on the Siberian Plain for both periods, where snow depths are relatively weak (< 0,30 m), for example on the Verkhoyansky mountains, where the large decline in snow depth during the summer months is well

represented. This performance has been already highlighted by Brun et al. (2013) with the previous Crocus-ERAI product in terms of low density, which is explained by the ability of Crocus to simulate the metamorphism of the snowpack layers into

depth hoar under extreme temperature gradients. Further west, in the East Siberian lowlands, deviations from observations occur at very low values. Wherever the relief is not significant, in the plain but around the mountain terrain, the observed snow depth compares reasonably well with in situ observations. This was highlighted by Mudryk et al. (2025) in its evaluation

of gridded SWE products for their ability to represent climatology, variability and trends in regions spanning the Northern Hemisphere.

The upper part of Figure 5 shows the spatial distribution of the time-averaged snow depth in early spring, as simulated by Crocus-ERA5 for the period 1950-2022. The figure depicts the long-term daily snow depth on 10 March for both the Canadian and Eurasian domains. The bottom panels show the results for a selection of stations representing diverse geographical regions.

The first Canadian station on Figure 5 (station S2) is situated at an elevated altitude (1323 m) in the vicinity of the Canadian Rocky Mountains. Others stations are located in the Canadian Shield region, which is characterised by numerous hills and glacier-carved lakes. The observed and simulated time series of snow depth on 10 March are in very good agreement over the Canadian Rocky Mountains for the present-day period (station S2), which is characterised by a high variability. In the northern continental part, between 65° and 70° latitude (stations S29 and S25), snow depths exhibit less variability around 0.4

m but also show excellent agreement with observations. In the northern Hudson Bay Lowlands (station S12), there is also a high degree of agreement between the two time series, although less agreement is revealed for the past climate (before 1980). Nevertheless, confidence in the Crocus-ERA5 snow depth in past climates is however supported by its close phase with in situ observations in the Arctic region (S37 and S38). However, in these very high-latitude regions, snow depths are overestimated by Crocus-ERA5, particularly in the north of Baffin Island which is divided into numerous peninsulas (S30).

In Eurasia (left panels of Figure 5), in the part of the Ural Mountains near the Kara Sea (station S19), time series of snow depth on 10 March show an overestimation of Crocus-ERA5 in the past climate (1960-1984) with a phase close to the observations. However, in the subsequent present period (1984-2012), Crocus-ERA5 and the in situ observations demonstrated a high degree of agreement. Near the city of Norilsk (station S28), Crocus-ERA5 exhibits three distinct behaviours: an overestimation for the period 1950-1967, followed by an underestimation for 1968-2000, ending with an excellent agreement in phase and

magnitude for 2000-2012. On the Central Siberian Plateau, in a station located in a latitudinal valley near the city of Toura (station S16, 277 m), Crocus-ERA5 overestimates snow depth but is remarkably in phase for the entire period of 1950-2015. This significant overestimation of Crocus-ERA5 in relation to observations associated to a remarkable agreement in phase is also shown in the data from the S8 station (682 m), located in the Kolyma mountain. In regions along the Laptev and Siberian Seas (stations S34 and S39), where snow depth varies below 0.5 m, Crocus-ERA5 exhibits a slight tendency to overestimate

observations. However, both the model and observations remain in relatively close alignment. Finally, there is a noticeable match in phase and magnitude to the west of the Kamchatka Peninsula (station S2), where deep snow depths generally exceed 1m. Figure 6 compares the Crocus-ERA5 and observations for the period 1950-2012 in terms of average duration, maximum, first and last day of continuous snow on the ground. A day with snow on the ground is defined as a day with more than 1 cm of snow (Brun et al., 2013; Decharme et al., 2019). Overall, Crocus-ERA5 demonstrates a satisfactory level of concordance with

the observations. This underlines the efficacy of the model in replicating the seasonal snow cycle in the Northern Hemisphere, despite the persistence of certain biases. Crocus-ERA5 simulates predominately a shorter snow season in the Arctic compared

**Table 2.** Statistics of biases between monitoring stations and Crocus-ERA5 for two levels of climatic normals: the period 1950-1980 related to the past climate and the period 1981-2012 related to the present climate.

| Dataset | 1950-1980 | | | | | 1981-2012 | | | | |
| | Cocus | Obs | | | | Cocus | Obs | | | |
| Variables | Mean | Mean | Bias | R | RMSE | Mean | Mean | Bias | R | RMSE |
| --- | --- | --- | --- | --- | --- | --- | --- | --- | --- | --- |
| averaged snow depth (m) | 0.07 | 0.07 | 0.01 | 0.82 | 0.06 | 0.07 | 0.08 | -0.00 | 0.84 | 0.06 |
| annual maximum snow depth (m) | 0.29 | 0.30 | -0.01 | 0.80 | 0.20 | 0.31 | 0.34 | -0.02 | 0.84 | 0.19 |
| date of maximum snow depth* | 197.84 | 199.72 | -1.88 | 0.55 | 35.24 | 198.34 | 200.25 | -1.91 | 0.60 | 32.30 |
| number of days with snow per year | 85.97 | 86.54 | -0.57 | 0.97 | 18.49 | 88.66 | 90.71 | -2.06 | 0.98 | 17.17 |
| first day of continuous snow* | 112.69 | 104.85 | 7.84 | 0.86 | 20.12 | 109.90 | 101.70 | 8.20 | 0.84 | 20.21 |
| last day of continuous snow* | 238.70 | 249.95 | -11.25 | 0.88 | 21.92 | 240.44 | 252.43 | -11.98 | 0.88 | 21.36 |
| duration of continuous snow | 78.44 | 75.19 | 3.25 | 0.96 | 23.48 | 80.99 | 78.74 | 2.25 | 0.96 | 22.08 |

*number of days since 1st August

to observations, while in the sub-Arctic plains it predicts a longer snow season. However, these biases are relatively small in both regions (~2 days, Figure 6.a). Two regions differ significantly from the observations. Around the Rocky Mountains, the Crocus-ERA5 show many more continuous snow days than observations. Additionally, in Norway, the biases alternate strongly between positive and negative values (~20 days). The biases in the first and last snowy days of the continuous snow period indicate that the snow season generally starts later and ends earlier in the Crocus-ERA5 estimates (Figures 6.c and 6.d). The largest discrepancies in snow peak between Crocus-ERA5 estimates and measurements are scattered in the Arctic region, where ERA5 has difficulty producing accurate estimates due to incomplete or sparse observational data. In the western part of Eurasia below 60° latitude, the bias in the maximum date of snowfall is small, approximately 2 days (Figure 6.b and Table 2), allowing accurate estimation and forecasting of changes in snow cover.

Table 2 shows the statistics of some Crocus-ERA5 variables related to the station measurements for the two periods 1950-1980 and 1981-2012. The lowest interannual correlations (R) are for the date of maximum snow depth with 0.55 and 0.60 for the periods 1950-1980 and 1981-2012 respectively, although this date remains quite close on average for both in situ observations and Crocus-ERA5, whatever the period. This point is relevant because of long-term trends in climate change may shift the maximum date of snowfall. Our results show that the climate has remained rather stable in the years leading up to 2012. All others variables exhibits significant correlations with R > 0.8. The strongest correlations (R>=0.96) are for the number of days with snow and the duration of continuous snow cover, reflecting the quality of both the Crocus snow model and the precipitation estimated by ERA5.

## 3.3 Snow Cover

The monthly SCE differences between Crocus-ERA5 and IMS anomalies, both calculated relative to the 2000-2022 baseline (Figure 7, left) were assessed in terms of their seasonality and interannual variability, as well as the climatology of each dataset (Figure 7, right). Although the differences are striking, with a high degree of interannual variability, three points stand out :

(i) an alternation between positive (violet) and negative (rust) SCE anomaly differences in autumn and winter (October-February), with positive values during the 2000s and negative values during the 2010s ;

(ii) a strong interannual agreement during the spring period (MAM) which is reflected in the overlap of the SCE climatological curves from Crocus-ERA5 and IMS, with March marking the key overlap point ;

(iii) an alternation between negative (rust) and positive (violet) SCE anomaly differences in June, opposite to that of autmn and winter.

Minor differences in small orders of magnitude, arising from shifts between positive and negative anomalies, are also evident.
The winter-scale shift in the Crocus-ERA5 and IMS relationship – positive anomaly differences in the 2000s (Crocus-ERA5>IMS) versus negatives ones in the 2010s (Crocus-ERA5<IMS) – likely reflects not only biases in both products but also changes in the representation of interannual variability. The anomaly differences are not driven solely by mean-state biases but also by the response of IMS to the evolving climate conditions. In particular, the sign reversal suggests a temporal shift in the relative representation of autumn-winter snow cover anomalies between Crocus-ERA5 and IMS. To simplify the discussion,
we define the anomaly difference as :

$$D = (SCE_C - clim_C) - (SCE_I - clim_I) = (SCE_C - SCE_I) - (clim_C - clim_I) \tag{1}$$

where, $C$ refers to Crocus-ERA5 and $I$ to IMS.

During warmer autumn-winter conditions in the 2010s, the anomaly differences are frequently negative ($D < 0$). This indicates that the deviation of Crocus-ERA5 from its own climatology is smaller than that of IMS. At the same time, the climato-
logical profiles show that Crocus-ERA5 generally has a slightly higher climatological SCE than IMS, so the negative cannot be explained by the baseline offset between the two datasets. Instead, it results from the way IMS classifies snow: its threshold-based detection makes it more sensitive to snow loss events under mild conditions. In other words, under warmer conditions IMS tends to detect snow loss more readily than Crocus-ERA5. In this case, IMS SCE departs more strongly from its own climatology, leading to larger negative anomalies relative to Crocus-ERA5. Consequently, IMS tends to record stronger SCE
negative anomalies relative to its climatology, whereas Crocus-ERA5, being physically constrained by energy and mass balance processes, shows a more gradual response. In practice, this means that under warm winters IMS anomalies are amplified relative to Crocus-ERA5.

In contrast, during spring (from May to June), the situation tends to be reversed and anomaly differences becomes more often positive ($D > 0$) in the 2010s. In this case, IMS tends to retain snow longer than Crocus-ERA5, resulting in overestimation
of snow cover relative to its climatology. This seasonal asymmetry –negative $D$ in autumn-winter and positive $D$ in spring

in warmer years – illustrates the contrasting behavior of the two datasets: IMS responds more abruptly to early-season snow loss but more conservatively to late-season snow retreat. It aligns with the documented tendency of IMS to overestimate SCE during spring melt, reflecting the vulnerability of its binary classification system to misclassifying thin or patchy snow as absent (Ingleby et al., 2024; Brown and Mote, 2009). Accordingly, the anomaly differences reverse sign relative to winter, with the opposition most striking in June. Since the physical configuration of Crocus-ERA5 is relatively stable over time, these shifts may instead point to methodological changes in the IMS analysis framework, consistent with documented evolutions in its analysis system (Helfrich et al., 2007; Estilow et al., 2015).

Figure 8 illustrates the evolution of the monthly anomalies of continental SCF across the Northern Hemisphere from Crocus-ERA5 and IMS for the period 2000-2022 . The correlation between the two datasets is 0.72, indicating they originate from the same distribution. Both time series show very similar seasonal variability, but there are differences in the magnitude of the peaks, which may explain the difference in trends. The long-term trend of SCF in the northern hemisphere from 2000 to 2022 is slightly downward in both time series although the Crocus-ERA5 exhibits a steeper decline compared to IMS with respectively a slope of $-1.9 \times 10^{-12}$ and $-4.5 \times 10^{-13}$. In physical terms, these slopes suggest a gradual trend in snow cover anomalies over 2000–2022, evolving slowly over very large timescales. This highlights the progressive nature of changes.

Snow cover exhibits strong variability depending on geographical location and local climatic conditions, particularly in northern regions where vegetation emerges in early summer. Because Crocus-ERA5 does not account for interactions with vegetation anomalies, it may misrepresent these dynamics, leading to an earlier and more rapid snowmelt compared to IMS. In addition, Crocus-ERA5 represents only open areas with low vegetation and therefore cannot fully capture wind-driven snow redistribution, introducing potential biases in snow depth and duration in regions where blowing snow and drifting are dominant processes. These limitations, however, must be considered together with those of the IMS binary classification system, which is generally robust during midwinter but more uncertain during transitional periods. Figure 9 compares the Northern Hemisphere-wide SCF from Crocus-ERA5 with the IMS long-term analysis to highlight these biases. During autumn (SON) and winter (DJF), above 60°N, Crocus-ERA5 and IMS show a strong agreement, as most of continental North America and Eurasia is entirely covered with snow. Conversely, south of 60°N in mountainous regions, Crocus-ERA5 shows a positive biais in SCF compared to IMS, likely due to its coarse topographic representation.

Below 60°N, in mountainous regions where snow cover is highly uneven, a positive bias is found between Crocus-ERA5 and IMS. This may be related to a poor representation of the topography at 0.25° resolution and thus to an overly simplistic relationship between simulated snow height and diagnosed snow cover used in Crocus in mountainous regions at this resolution. Snow extent seasonality is closely related to air surface temperature and radiative forcing, and in autumn and winter (low radiative forcing) is influenced by both temperature and precipitation. On the plains, and with accurate atmospheric forcing, Crocus-ERA5 shows excellent performance.

During spring (MAM), snow thickness decreases as temperatures (and radiative forcing) increase, leading to snow thinning and eventual melting. The spring thaw typically starts at lower latitudes and elevations and gradually moves northwards and to higher elevations as the season progresses. In Crocus-ERA5, snow melts more rapidly than in IMS over northern western Canada, southern eastern Canada and eastern Siberia. This bias is partly due to the model's lack of representation of boreal

forests at high latitudes (below 60°N), which delay snowmelt by limiting incoming radiation at the snow surface (Decharme et al., 2019). The presence of the boreal forest prolongs the duration of snow cover in comparison to areas devoid of such vegetation. In Crocus-ERA5 the snowpack is simulated as an open-field surface so it does not take into account such process. Moreover, Crocus-ERA5 assumes a fixed grassland, so it cannot capture impact of tundra emergence above 60°N, which alter snowpack dynamics in early summer.

In summer (JJA), the negative effect of high altitude on the agreement between the two snow cover products is reduced by the absence of snow. There are negative biases in the northernmost part of the Arctic, excluding Greenland, which correspond to the largest region of the tundra biome, however the differences remain below 10%.

In autumn (SON), the emergence of continental snow cover shows significant bias only over mountains range as Yablonoi and Rocky Mountains.

## 4    Conclusions and Perspectives

This paper is based on the two most representative variables of snow characteristics and commonly used to monitor snow, snow depth and snow cover. The challenge was to validate the model simulations against independent data in order to minimize the uncertainty attributed to the interdependence between the explained and independent variables. Strengths of this study include direct comparisons of Crocus-ERA5 with long-term homogeneous time series from in situ observations snow depth and gridded satellite multi-sensor analyses of snow cover fraction. Comparisons reveal an excellent overall performance of Crocus-ERA5 in capturing snow dynamics in the Northern Hemisphere.

Sensitivity to atmospheric forcing was assessed by comparing Crocus-ERA5 with Crocus-ERA-Interim (Crocus-ERAI), for the common period 1979 to 2018. ERA5 forcing uses an advanced land data assimilation system. Its higher spatial resolution of 0.25° provides more detailed representations of topography and land cover than ERAI, which has a resolution of 0.75°. This led to an increase in snow depth and a decrease in snow cover in Crocus-ERA5 compared to Crocus-ERAI. Regarding snow depth, Crocus-ERA5 is in agreement with Crocus-ERAI in terms of interannual variability and seasonal cycles in Eurasia and North America. It is noteworthy that Crocus-ERA5 tends to produce stronger anomalies, which are consistent with improvements in the atmospheric forcing. A persistent negative trend in snow depth over Eurasia is evident since 2000, while North America shows greater interannual variability. The statistics (mean and standard deviation) show a shift in the Crocus-ERA5 distribution toward higher values. The standard deviation varies similarly around the mean for both Crocus-ERAI and Crocus-ERA5 distributions, in both regions, Eurasia and to a lesser extent, North America. Crocus-ERA5 has generally higher snow depth than Crocus-ERAI in spring, especially in Eurasia.

The climatological periods 1950-1980 and 1981-2012 reveals strong agreement in snow depth between Crocus-ERA5 and in-situ data. The ability of Crocus-ERA5 to represent daily snow depth (or mass, related to depth by density) is assessed relative to long-term snow depth on March 10 for various in-situ stations covering different geographic regions of Canada and Eurasia. The Crocus-ERA5 long-term snow depth aligns remarkably well with in-situ observations, demonstrating its ability to capture interannual variability. In Canada, the simulation performs well across in a variety of environments, including complex regions

such as mountains, hills and glacier-carved lakes. In Eurasia, although there is a slight overestimation of snow depth in earlier decades, Crocus-ERA5 shows good phase and amplitude agreement in snow occurrence, particularly from the 1990s onward.

Compared to the IMS multi-sensor product, the differences between monthly snow cover extent (SCE) anomalies for Crocus-ERA5 and IMS, reveal that they arise primarily from the threshold-based snow classification used in IMS, which under warmer autumn-winter conditions amplifies snow loss relative to the more physically constrained Crocus-ERA5. In early summer, Crocus-ERA5 tends to underestimate SCE because of its idealized grassland surface with fixed climatological properties and its omission of forest-snow interactions. This simplification introduces a bias, causing premature snow loss in regions where vegetation is evolving. For example, Crocus-ERA5 fails to capture the impact of tundra emergence at higher latitudes (above 60°N). In contrast, IMS often overestimates the SCE by misclassifying thin or patchy snow as continuous cover. These opposing biases partly offset each other, but also exacerbate the discrepency between Crocus-ERA5 and IMS. Consequently, CROCUS-ERA5 provides reliable snow dynamics over most months and regions.

To effectively assess water resources and forecast spring runoff, it is essential to measure the total amount of water stored in the snowpack using SWE as a key indicator. SWE reflects snow density and compaction and plays a critical role in Arctic amplification and climate change by influencing the timing and extent of snowmelt. There are major differences in temporal and spatial resolution between existing SWE products (Mudryk et al., 2025), which significantly limits their usefulness in cryosphere and climate change studies. Long-term time series from climate models or remote sensing data have often limited spatial accuracy, especially in areas where snow processes, such as melting and compaction, are not well represented. Winkler et al. (2021) proposed a method for estimating SWE using only snow depths, which outperforms models based on empirical regressions. Fontrodona-Bach et al. (2023) regionalized this method to create the NH-SWE dataset for the Northern Hemisphere. Shao et al. (2022) developed a high accuracy SWE product by integrating various existing SWE data sources into a Ridge Regression Model (RRM), using machine learning. The temporal resolution of the RRM SWE product is daily, and the spatial resolution is 10 km. The study demonstrated the effectiveness of this method for creating global SWE products , providing consistent seamless spatial and temporal coverage. The RRM SWE product minimizes dependence on a single SWE dataset by optimally leveraging multiple SWE sources and taking altitude into account. Although this paper focuses on direct spatio-temporal validation of the Crocus-ERA5 snow product using independent data, an interesting future direction would be to evaluate the Crocus-ERA5 SWE against these new datasets. It should be noted, however, that recently Mudryk et al. (2025) showed that Crocus-ERA5 was one of the most effective product for reproducing SWE in the Northern Hemisphere, at least in plain areas.

## 5 Data availability

The new *Crocus-Era5* dataset is free to access and available at https://doi.org/10.5281/zenodo.14513248 (Decharme et al., 2024). The dataset is provided over the period 1950-2023 in netcdf format and contains modelled daily snow depth, snow water equivalent, liquid water content in the snowpack, snow albedo, snow surface temperature, snowpack internal temperature, and snow cover fraction. The previous *Crocus-Era-Interim* dataset is free to access and available at https://doi.org/10.5281/zenodo.

14513040 (Decharme, 2024). The dataset is provided over the period 1979-2019 in netcdf format and contains modelled daily snow depth and snow water equivalent, as well as monthly snow surface temperature, snowpack internal temperature, and snow cover fraction.

570 *Author contributions.* BD defined the scientific framework, performed the numerical simulations and supervised the findings. SRB defined the methodology and selected the relevant datasets, pre-processed the observational data, developed the analytic calculations, performed the analysis and wrote the manuscript. AB and LF performed the ERA5 and the ERA-Interim atmospheric forcing, respectively.

*Competing interests.* The authors declare that they have no competing interests.

*Acknowledgements.* We acknowledge the National Snow and Ice Data Center for making the Interactive Multisensor Snow and Ice Mapping
575 System (IMS, U.S. National Ice Center 2008) data available and making this comparison possible.

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

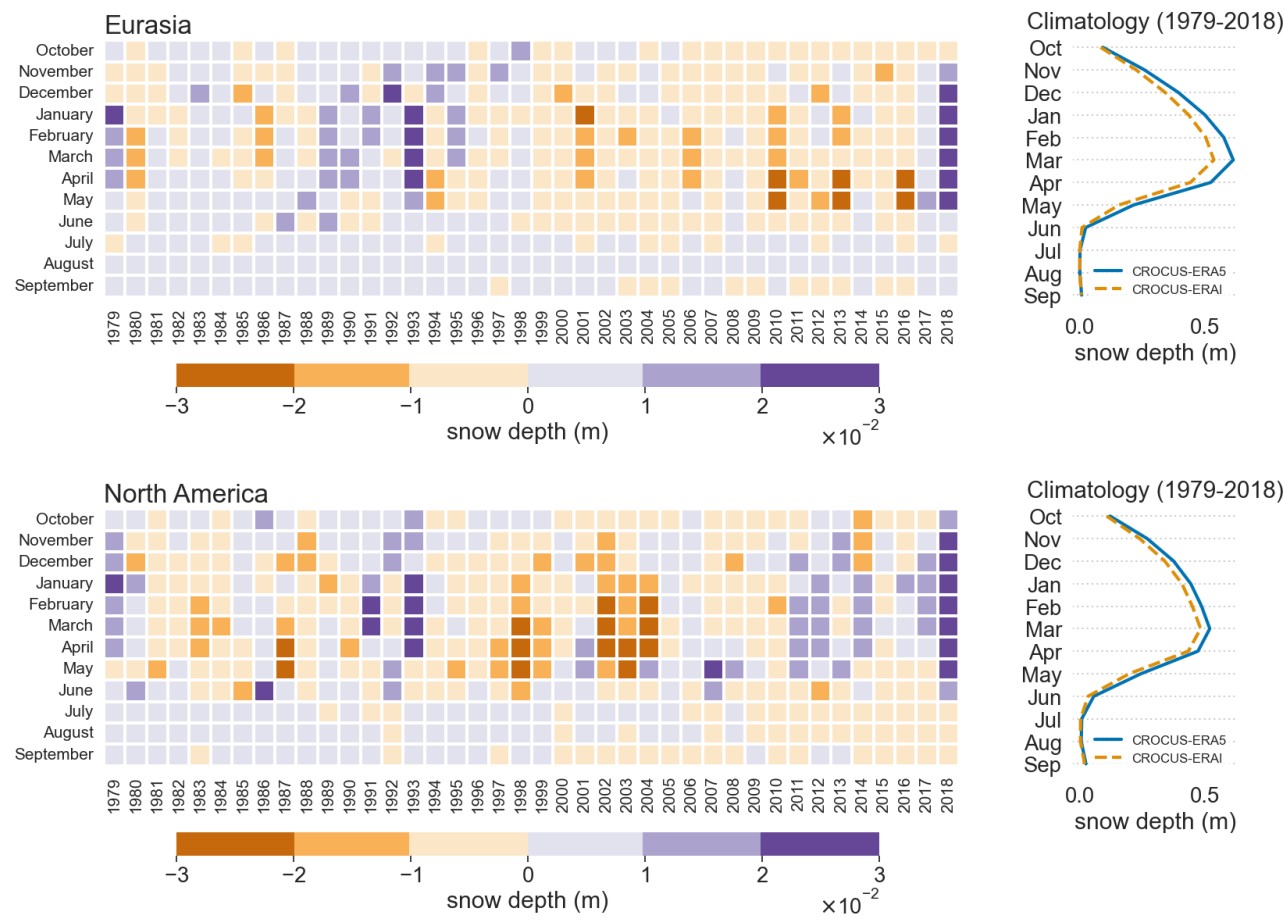

**Figure 1.** (Left) Temporal evolution of the difference between individual snow depth anomalies (Crocus-ERA5 - Crocus-ERAI), with months on the y-axis (October at the top, following the hydrological year) and years on the x-axis (1979–2018). Color indicates the magnitude of the anomaly difference, highlighting both intra-annual (seasonal) and interannual (year-to-year) variability. (Right) Monthly climatology of snow depth from Crocus-ERA5 (blue line) and Crocus-ERAI (dashed orange line). The x-axis shows climatological values, and the y-axis lists months from October (top) to September (bottom).

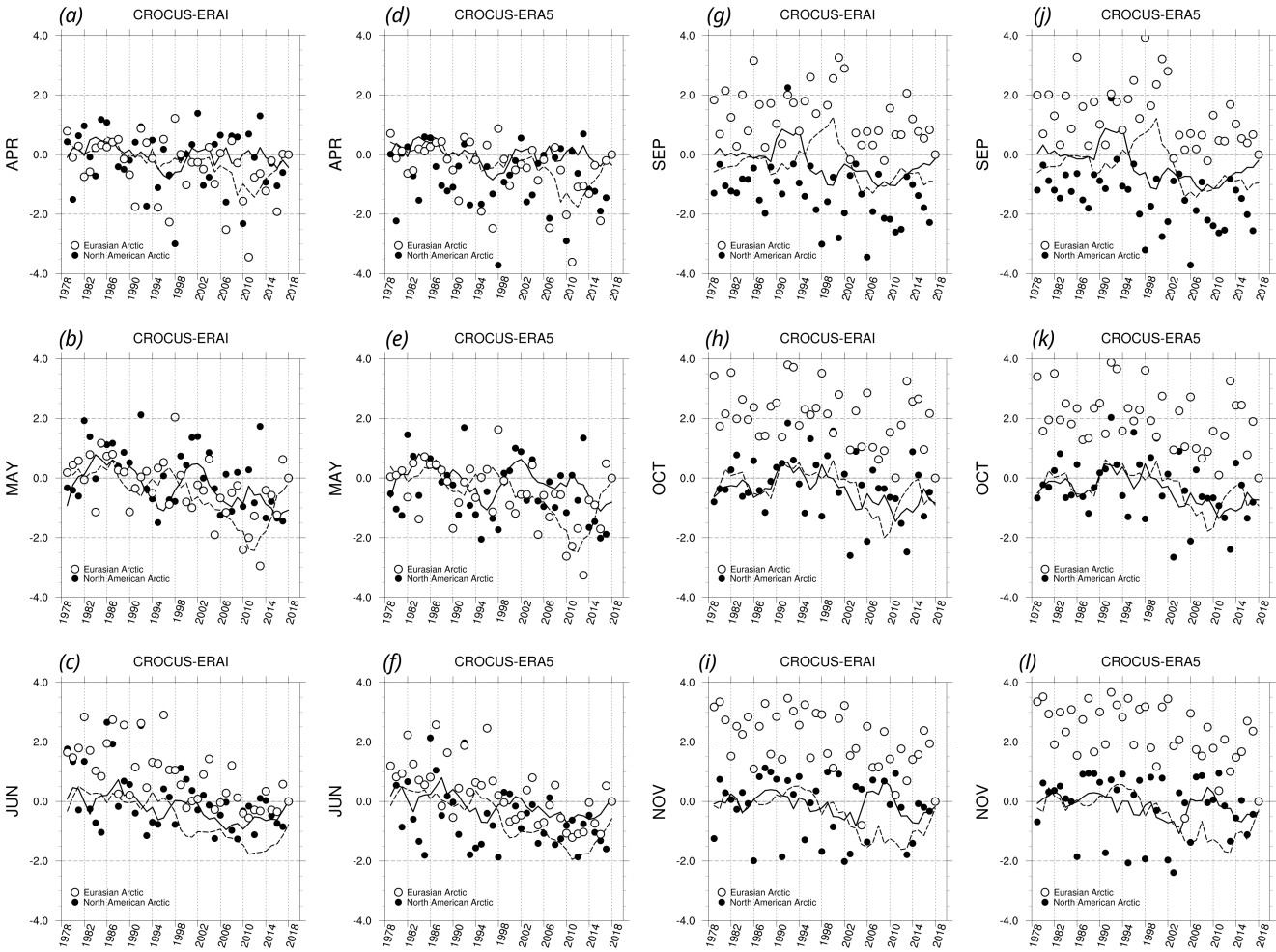

**Figure 2.** Monthly time series of snow cover extent (SCE) shown as standardized anomalies relative to the period 1979-2018. Spring and Autumn are represented respectively for months of April-May-June, from (a) to (f), and September-October-November, from (g) to (l). Land surface averages for North America and Eurasia refer to latitudes above 60° N. Solid black and dash lines depict 5-yr running means providing a smoothed trend of variability. Circles are used to highlight anomalies relative to the last year 2018. Adapted from Figure 1 in Mudryk et al. (2024).

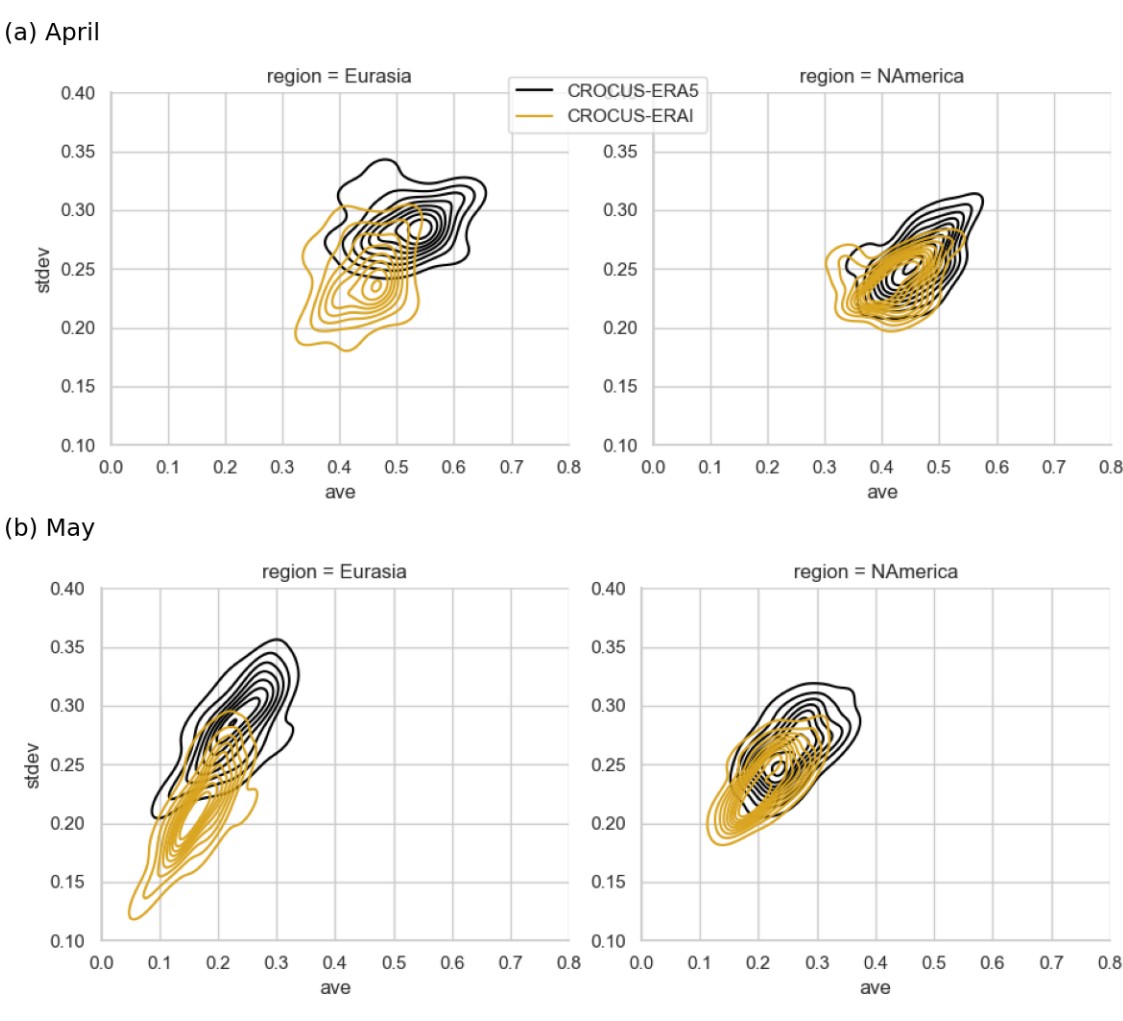

**Figure 3.** Monthly standard deviation of snow depth from its mean on the land surface for Eurasia (left) and North America (right) with respect to latitudes above 60° N. Representation of the months **(a)** April and **(b)** May.

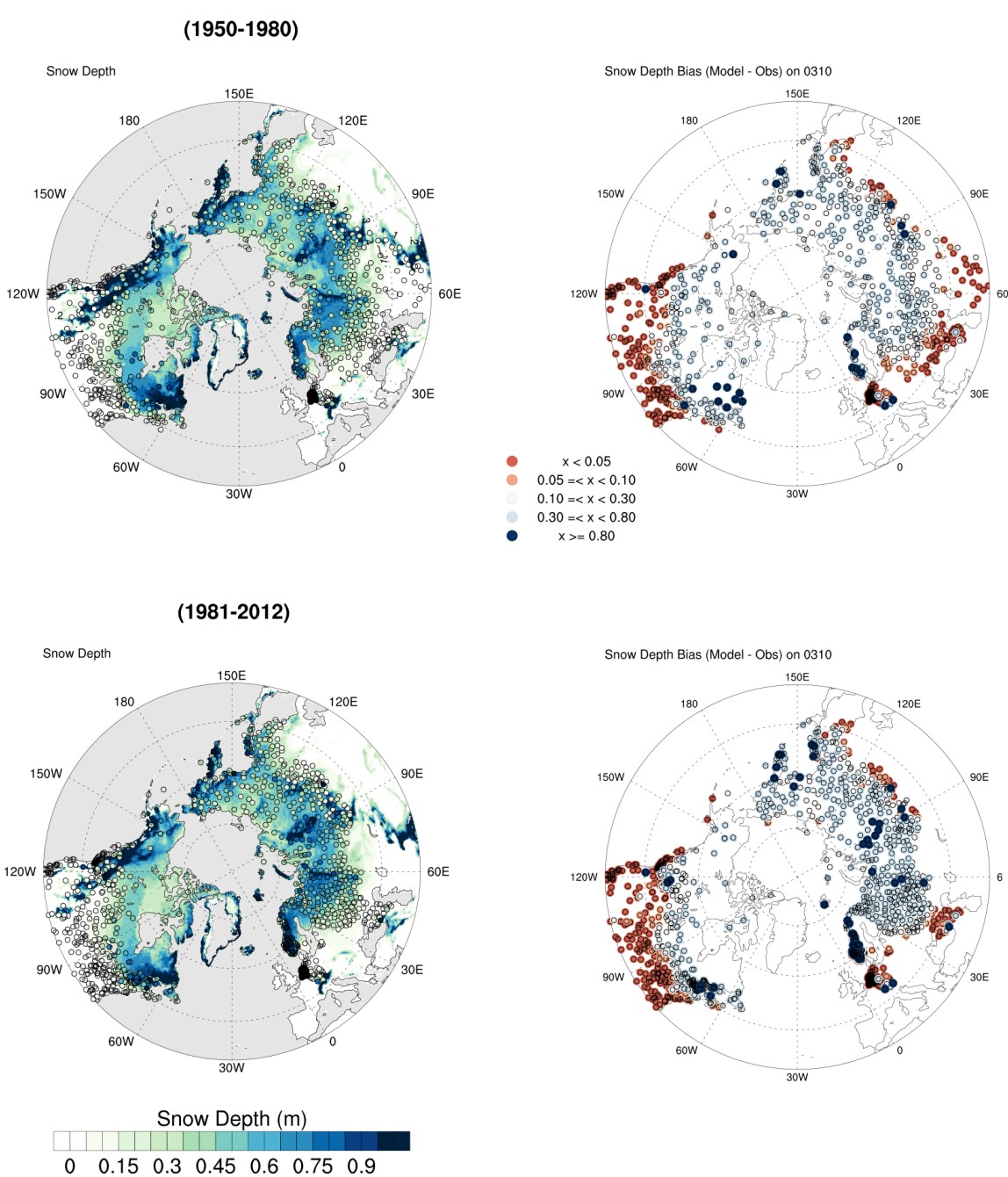

**Figure 4.** (Left) Time-averaged snow depth from Crocus-ERA5 (2D field) for two climatological periods: 1950–1980 (top) and 1981–2012 (bottom), both on 10 March. Observed snow depth is shown as colored circles, using the same color scale as the background field. Visual interruptions in the field indicate discrepancies between the model and observations. (Right) Corresponding biases (model minus observation) for each period.

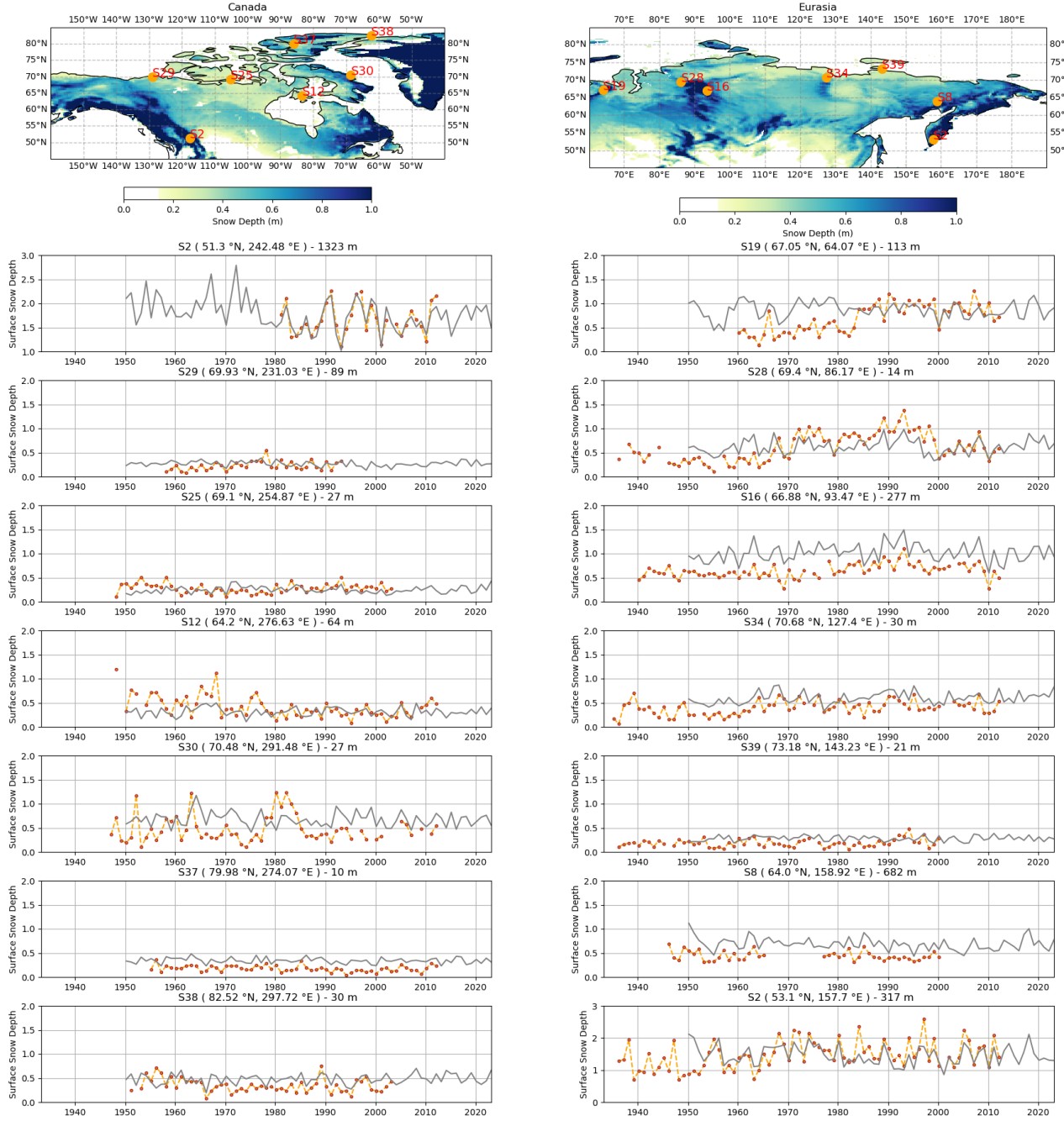

**Figure 5.** (Top) Location of observational stations over the spatial distribution of time-averaged snow depth in March (1950–2022), for two domains: Northern North America (left, stations located in Canada) and Eurasia (right). (Bottom) For each domain (column), time series of snow depth on 10 March are shown for each station, comparing Crocus-ERA5 (grey lines) with observations (orange circles connected by dotted lines). Each time series panel corresponds to a station marked on the map above. The x-axis starts at the beginning of the longest available time series.

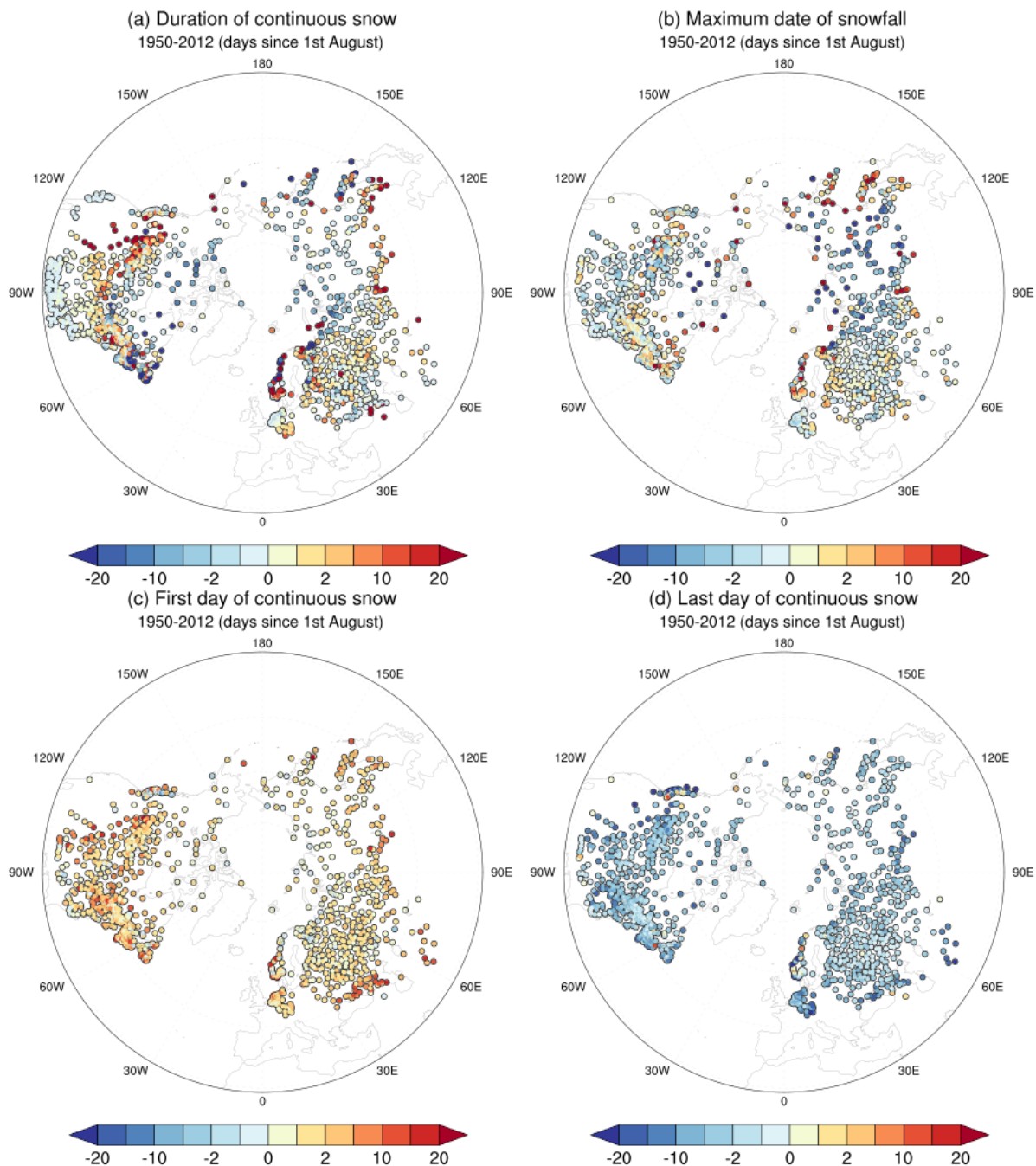

**Figure 6.** Bias between measurement stations and Crocus-ERA5 for 1950-2012 concerning their temporal phases: (a) Duration of continuous snow, (b) Maximum date of snowfall, (c) First day of continuous snow and (d) Last day of continuous snow.

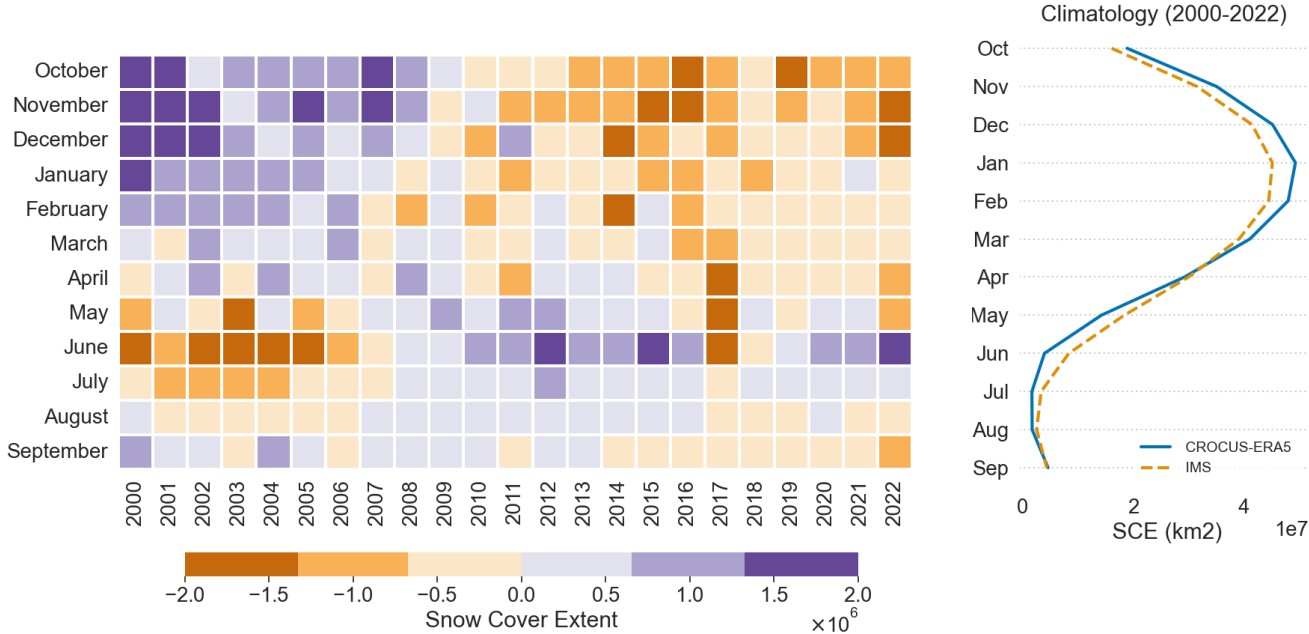

**Figure 7.** (Left) Temporal evolution of the difference between individual snow cover extent anomalies, with months on the y-axis (October at the top, following the hydrological year) and years on the x-axis (2000–2022). Colors indicates the magnitude of the anomaly difference, highlighting both intra-annual (seasonal) and interannual (year-to-year) variability. (Right) Monthly climatology of SCE from Crocus-ERA5 (blue line) and IMS (dashed orange line). The x-axis shows climatological values, and the y-axis lists months from October (top) to September (bottom).

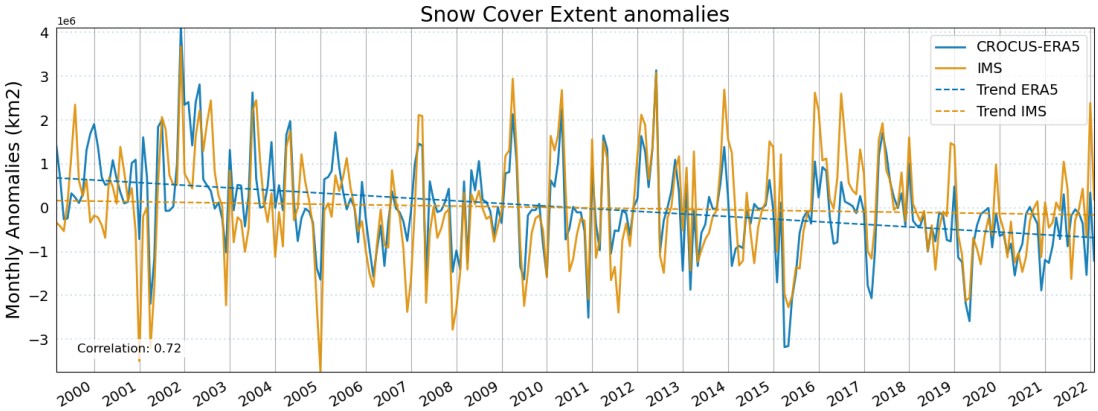

**Figure 8.** Time series of monthly SCE anomalies from January 2000 to December 2022, shown as a continuous line. Dashed lines indicate the linear trends over the period.

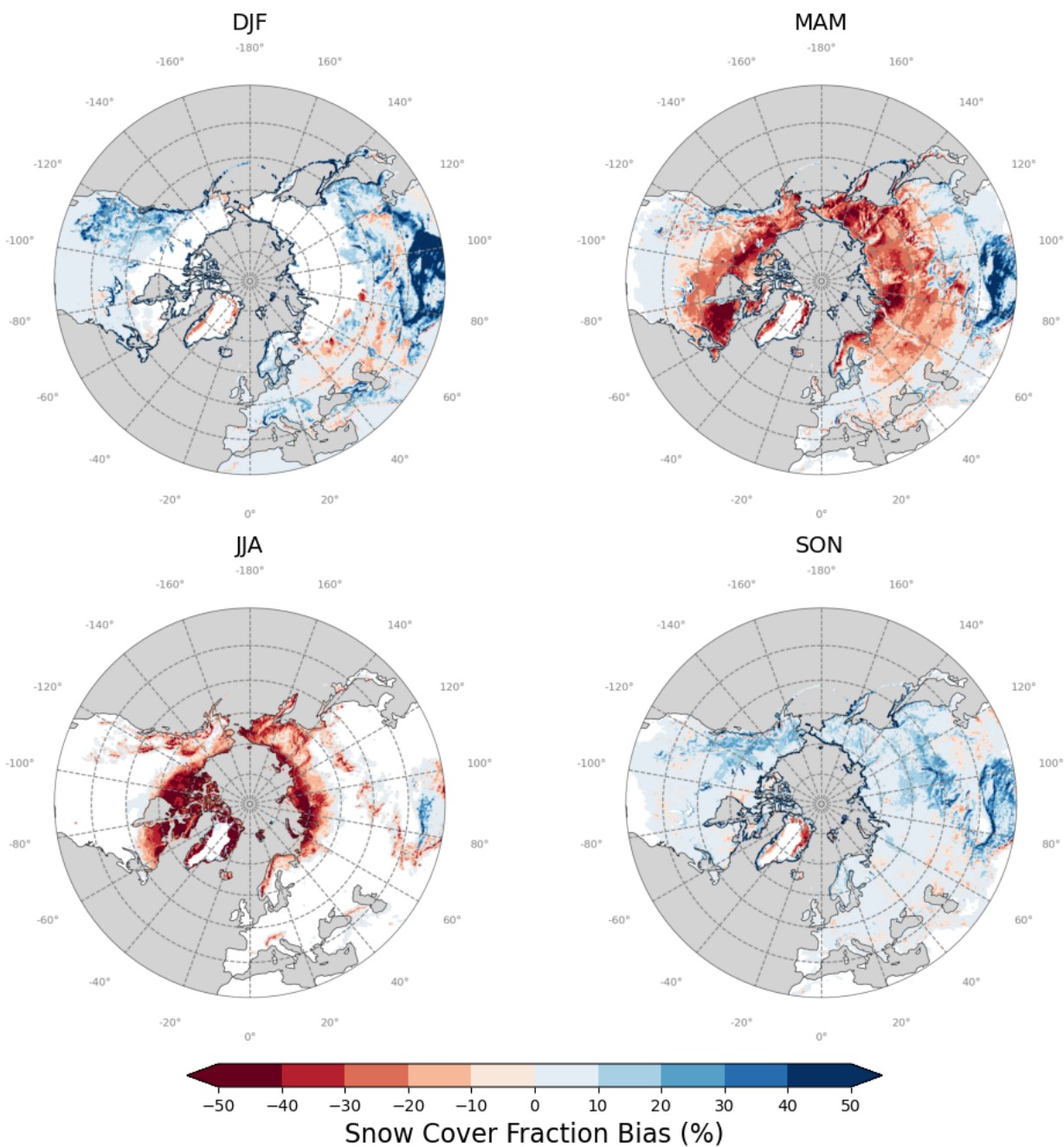

**Figure 9.** Seasonal mean snow cover fraction (SCF) biases of Crocus-ERA5 relative to IMS estimates over the Northern Hemisphere for the period 2000-2022. Panels show spatial distributions of SCF bias (model minus observation) for each season: winter (DJF), spring (MAM), summer (JJA) and autumn (SON)..