# Peer review of "Insights into the North Hemisphere daily snowpack at high resolution from the new Crocus-ERA5 product"

_Earth System Science Data, 2024_

## Referee Comment (RC2)

**April 7th, 2025**

Insights into the North Hemisphere daily snowpack at high resolution from the new
Crocus-ERA5 product
By: S. Ramos Buarque et al.

**General Comments**

This paper compares Northern Hemispherical snow datasets (snow depth and SCF) from
Crocus-ERA5 to long-term in situ station observations and its predecessor model,
Crocus-ERA-Interim. It is important to assess the accuracy of this dataset, as both CNRM and
ECCC are using it to evaluate changes in Arctic snow cover.

I believe Reviewer 1's comments addressed many of the concerns I had with the manuscript, and
the authors' subsequent responses have led to marked improvements. Therefore, I have taken
those comments and improvements into consideration in this review.

While this paper has the potential to be a strong contribution to the literature, the manuscript
requires significant restructuring. There is a lot of valuable information and analysis, but it is
scattered somewhat randomly throughout the sections, making it challenging to follow exactly
what the authors did. This is especially important for a paper submitted to *ESSD*, as the journal
primarily focuses on dataset production and methodology.

I believe the updated analysis provided by the authors is robust, so my comments mainly focus
on providing more detail on the datasets/models and reorganizing Sections 1–3. That said, I
believe these changes constitute a major revision, as large portions of text should be moved and
better integrated into the appropriate sections.

**Specific Comments**

1.  Introduction

    L49–65: As described in more detail below, remove this text and integrate into Section 2.

2.  Section 2. Data/Methods: This section contains a mix of background information, data
    description, and is confusing to read. For example L80-83, the section begins with a topic
    sentence that is background information that has limited pertinence to the data or
    methods described in this paper:

*"Warming and more frequent winter thaws are contributing to changes in snow pack structure with important implications for snow distribution. The performance of snowpack modelling in this context of climate change, can be summarized by the two main variables used as indicators of climate change because of their interactions and feedbacks with surface energy: the snow depth and the snow cover."*

This is a well written passage, but belongs in the Introduction!

Given this paper being submitted to ESSD, I believe there needs to be a central description of the modeling framework in the Data section. This information could be moved and updated from L49-65. Some questions to think about:

How does Crocus work (i.e., model physics)?
What variables does it need to run?
What variables does it produce?
How are Crosus-ERA5 and Crocus-Interm-ERA5 different?

Some of this information is in the paper but scattered in many different places. Please include a table that clearly shows the difference between Crosus-ERA5 and Crocus-Interm-ERA5 differences (i.e., spatial resolution, time period, model physics, etc.).

I'm not going to go line by line here, but have one subsection describing the models/data used in this study and one subsection describing the methods used to analyze these data. Move all other information that justified the importance of modeling to the intro. Again, I think most of this text is very well written, just needs to be reorganized!

3. L139–148 (and new text added in response): This passage is a great data description. I think it belongs in the Data description section.

4. Figure 2: Updated figure from response document is much improved and addresses all my concerns with the original figure.

5. Figure 4: Change color scheme – this looks like an elevation model. White/gray to dark blue commonly used for snow depth.

6. Figure 5: Use the same color scheme as Figure 4 to keep consistent.

7. L245: Any statistical analysis should be described in the Methods section and not be first introduced here. As noted prior, I would create a methods subsection where all this information can be contained.

8. L352: Please include a paragraph summarizing the findings. The manuscript abruptly ends with discussion of future work.

**Technical Comments:**

Numerous technical and grammatical errors exist in the document. Please check for correctness closely. I only list a few below.

1. L259: "...reproduced well…"
2. L334: SWE already defined.
3. L339: Check citation.

---

## Author Comment (AC2)

**#essd-2024-451 - Author's response**

The authors thanks the reviewer for their constructive feedback and for recognizing the improvements made following the initial round of revisions. We appreciate the acknowledgement that the updated analysis is robust and that the paper has potential to be a strong contribution. Suggestions contributed to clarifying key aspects of the manuscript and improve presentation of our results in line with ESSD's focus.
* * *
**Reviewer 2**

**General Comment** — This paper compares Northern Hemispherical snow datasets (snow depth and SCF) from Crocus-ERA5 to long-term in situ station observations and its predecessor model, Crocus-ERA-Interim. It is important to assess the accuracy of this dataset, as both CNRM and ECCC are using it to evaluate changes in Arctic snow cover.

I believe Reviewer 1's comments addressed many of the concerns I had with the manuscript, and the authors' subsequent responses have led to marked improvements. Therefore, I have taken those comments and improvements into consideration in this review.

While this paper has the potential to be a strong contribution to the literature, the manuscript requires significant restructuring. There is a lot of valuable information and analysis, but it is scattered somewhat randomly throughout the sections, making it challenging to follow exactly what the authors did. This is especially important for a paper submitted to ESSD, as the journal primarily focuses on dataset production and methodology.

I believe the updated analysis provided by the authors is robust, so my comments mainly focus on providing more detail on the datasets/models and reorganizing Sections 1–3. That said, I believe these changes constitute a major revision, as large portions of text should be moved and better integrated into the appropriate sections.

**Reviewer Comment 2.1** — Introduction
L49–65: As described in more detail below, remove this text and integrate into Section 2.

**Reply**: First of all, we would like to clarify that the changes were made to the text that had already been modified in response to Reviewer 1. The structure of the text has changed considerably. Several paragraphs have been moved to Section 2 (Data and Methods). These large-block moves make point-by-point explanations unhelpful. For this reason, we do not provide detailed explanations here. For example, lines 49–65 were split and redistributed into the Introduction (lines 49–51), Section 2.2 (lines 51–54), Section 2.4 (lines 56–57), and so on.

To track these changes, we used the trackchanges package in LaTeX. However, please note that when blocks of text were moved, the "old" locations are not highlighted. Only new insertions

appear as tracked changes. Due to limitations in LaTeX's ability to track changes across large text blocks, some modifications may not be fully visible. Nevertheless, all essential updates are retained in the LaTeX tracked version and can be seen in the diff/track PDF file.

**Reviewer Comment 2.2** — Section 2. Data/Methods: This section contains a mix of background information, data description, and is confusing to read. For example L80-83, the section begins with a topic sentence that is background information that has limited pertinence to the data or methods described in this paper:

"Warming and more frequent winter thaws are contributing to changes in snow pack structure with important implications for snow distribution. The performance of snowpack modelling in this context of climate change, can be summarized by the two main variables used as indicators of climate change because of their interactions and feedbacks with surface energy: the snow depth and the snow cover."

This is a well written passage, but belongs in the Introduction!

Given this paper being submitted to ESSD, I believe there needs to be a central description of the modeling framework in the Data section. This information could be moved and updated from L49-65. Some questions to think about:

How does Crocus work (i.e., model physics)?
What variables does it need to run?
What variables does it produce?
How are Crosus-ERA5 and Crocus-Interm-ERA5 different?

Some of this information is in the paper but scattered in many different places. Please include a table that clearly shows the difference between Crosus-ERA5 and Crocus-Interm-ERA5 differences (i.e., spatial resolution, time period, model physics, etc.).

I'm not going to go line by line here, but have one subsection describing the models/data used in this study and one subsection describing the methods used to analyze these data. Move all other information that justified the importance of modeling to the intro. Again, I think most of this text is very well written, just needs to be reorganized!

**Reply**: In response to the imperative need for clarification, we have classified and grouped the information as much as possible within Sections 1–3. We have carefully revised the manuscript with the goal of improving its clarity and organization, aligning it with ESSD's emphasis on dataset production and methodology.

The following improvements were made to enhance the clarity, structure, and readability of the manuscript:

- The introductory material has been reorganized to better guide the reader through the context.

- We have clarified the presentation of the datasets and provided a more detailed description of the Crocus model.

- Background information and methodological details have been consolidated and stream-lined.

- Redundant or scattered content has been moved to appropriate sections to enhance the flow and coherence of the manuscript.

The section 2 ("Data and Methods") now includes five subsections: Atmospheric Forcings, Crocus-ERA5 Framework, Snowpack Modeling, Analysis Methods, and Observational Data. In this section, we provide detailed responses to the reviewer's questions:

- **How does Crocus work (i.e., model physics)?**
  This is addressed in Section 2.3 *Snowpack Modeling*, where we describe Crocus as a multi-layer snow model.

- **What variables does it need to run?**
  Section 2.2 *Crocus-ERA5 Framework* specifies the input variables required to force Crocus, such as air temperature, humidity, wind speed, incoming shortwave and longwave radiation, and precipitation.

- **What variables does it produce?**
  Section 2.2 *Crocus-ERA5 Framework* lists the availables output variables, which are accessible at its Zenodo repository.

- **How are Crocus-ERA5 and Crocus-Interim-ERA5 different?**
  This is explained in Section 2.1 *Atmospheric Forcings*, which describes the differences between the ERA5 and ERA-Interim reanalysis used as atmospheric forcing for Crocus.

A table summarizing the differences between the ERA-Interim and ERA5 reanalyses, which are used to force Crocus, has been added.

**Reviewer Comment 2.3** — L139–148 (and new text added in response): This passage is a great data description. I think it belongs in the Data description section.

**Reply**: We appreciate the suggestion to move this section to section 2 ("Data and methods"). Now, this paragraph take part of Section 2.1 *Atmospheric Forcing* but were splited as its main part were putted in the Table 1 describing differences between reanalysis. All informations are preserved and enriched by highlighting the type of snow model used by the two reanalyses, which is single-layer and therefore takes no account of vertical variations in the snowpack.

**Reviewer Comment 2.4** — Figure 2: Updated figure from response document is much improved and addresses all my concerns with the original figure.

**Reply**: Figure 2 has been redrawn. It integrate now the autumn season (SON). This inclusion allowed us to better support the meaning of this representation concerning especially filled circles.

**Reviewer Comment 2.5** — Figure 4: Change color scheme – this looks like an elevation model. White/gray to dark blue commonly used for snow depth.

**Reply**: Thank you for the suggestion. The color scheme of Figure 4 has been updated to a white/gray to dark blue gradient to better represent snow depth.

**Reviewer Comment 2.6** — Figure 5: Use the same color scheme as Figure 4 to keep consistent.

**Reply**: The color scheme of Figure 5 has been updated to match Figure 4, ensuring consistency across the figures.

**Reviewer Comment 2.7** — L245: Any statistical analysis should be described in the Methods section and not be first introduced here. As noted prior, I would create a methods subsection where all this information can be contained.

**Reply**: We thank the reviewer for this suggestion. We have moved the description of all snow depth diagnostics to the *2.4 Analysis Methods* subsection. In the Section 3, these diagnostics are now only referenced without detailed definitions.

**Reviewer Comment 2.8** — L352: Please include a paragraph summarizing the findings. The manuscript abruptly ends with discussion of future work.

**Reply**: Thank you for the comment. The main findings have been summarized in the first part of the Conclusions and Perspectives section. We have reviewed and ensured that the key results are clearly presented before discussing future work.

**Technical Comments:**

Numerous technical and grammatical errors exist in the document. Please check for correctness closely. I only list a few below.

1. L259: "...reproduced well..."
2. L334: SWE already defined.
3. L339: Check citation.

**Reply**: Thank you for your careful review. We have re-read the manuscript carefully and had it cross-checked by co-authors as well. Regarding the specific points:

- L259 : The wording around has been revised for clarity.

- The abbreviation SWE (snow cover extent) was defined when it first appeared in the abstract, main text and each figure.

- Citations has been checked and corrected if necessary.

**Please note that Figures 7 and 8 have been updated in this version. Initially, we provided a rationale for the representation of the standardized snow cover extent anomaly. However, in light of Reviewer 1's comments and to ensure greater consistency across the dataset presentation, we have revised these figures accordingly. We hope these changes address the reviewer's concerns and enhance the overall clarity and impact of the manuscript."**

---

## Referee Report (RR1)

As noted in the original review, this paper can provide an important contribution; however, many of the key concerns expressed in the first round of review remain. Importantly concerns regarding how to meaningfully visualize results remain and there is a substantial lack of supporting citations throughout. Specific comments are included below.

Introductory paragraphs require supporting citations (e.g., P2 and P3 have minimal citations).

Section 2, starting at L75: arguments made in this section should be strengthened with supporting citations. This sentiment should be considered throughout the article.

L133: Since the early 2010s not "2010"

Section 2.3: More detail on the calculation of evaluated variables is valuable (e.g., mathematical representation of a snow depletion curve used to calculate snow cover extent).

Section 2.5: There is no guarantee that the in-situ snow depth observations, particularly with these observations tending to be in clearings, provide spatial representation allowing a fair comparison against simulation snow depth at a course (0.25-degree) resolution. There needs to be a qualification of this point. The elevation difference was considered, which is good, but other heterogeneities (e.g., from land cover, aspect, wind redistribution, etc) can play important roles.

IMS has known uncertainties and this should be addressed.

L230: The claim that forests do not have an important impact on interannual variability needs to be supported, proven, or removed. If removed, this limitation needs to be addressed.

L244: "Negative" rather than "Negatives"

Figure 1: In my original comment I had requested anomaly difference plots, rather than only a presentation of the anomalies. This point remains important to visualize the differences between ERA5 and ERAI. Showing the anomalies, as currently done, is valuable for the reason the authors' noted in their original response, but visualizing differences is also very important to understand discrepancies between the 2 versions. I disagree with the authors' argument on maintaining the calendar year ordering, as noted in their response, the snow accumulation starts in Oct-Nov (i.e., the start of the water year), so it would make sense to present results in this manner. This note applies also to Figure 7.

More clarification is needed for the presented anomaly calculation. For example, in panel 3 in 1995 there are negative snow anomalies in typical high SWE months (April-June) whereas SWE anomalies are positive in summer months when SWE tends to be minimal (Jul-Sept). In the calculation of the anomalies, are the respective monthly averages subtracted or the total time series average? If the latter, this result would suggest that SWE is relatively higher in summer months which would be perplexing. Thus, I assume the former is the case here which should be articulated.

Figure 2: Snow cover area, which is often represented as snow cover fraction in comparisons, is physically meaningful (e.g., for surface albedo) and allows comparisons across regions of different sizes. I maintain that showing the snow cover fraction here would be meaningful and would be a valuable supplementary figure.

The caption notes "Filled" circles, but the circles are empty.

In the original response, the authors' note that the figure format is selected to maintain consistency with prior studies to allow easy comparisons, which makes sense, however these 2 studies do not seem to be cited in this article.

Figure 4: I maintain that showing a difference plot is highly valuable here. If the authors' worry that this would add too much complexity to this figure, then the difference can be shown in supplementary. The difference would be highly insightful to present observed vs. modeled differences in snow depth between climatological periods.

There is also a key issue with the current representation in Figure 4. Namely, it seems the goal with this figure is to show whether the model captures the spatial distribution of snow depth relative to observations, but in areas with overlapping circles that are filled with color, there is no way to see the underlying observation color in the map. Therefore, it would be appropriate to present a bias map of the circles (small enough to reduce overlap) colored by the model bias (e.g., using a polar color scheme from red to blue to present underestimates to overestimates).

Figure 7: I largely disagree with the authors' argument to not show SCF, which I consider potentially more informative than a comparison of anomalies due to the physical meaningfulness of SCF. Furthermore, an anomaly comparison will mask systematic biases which are important to present. If the authors would like to present standardized anomalies, this is reasonable, but the physically meaningful comparison of SCF, which has the ability to provide a presentation of systematic biases, should be included as well (e.g., in supplementary). The SCE comparison is useful but has lower granularity.

The original suggestion of showing the difference panel remains, as this is needed to easily visualize the discrepancies between the model and observations.

Figure 9: The color bar label should note snow cover biases, rather than snow cover. A polarized color bar here is appropriate.

In the original response the authors' note that a detailed assessment for key snow-related variables in this data set (e.g., SWE and snow albedo) are beyond the scope of this study. Yet the opening sentence in the abstract is: "This article provides a detailed analysis of the Crocus-ERA5 snow product covering the Northern Hemisphere from 1950 to 2022". This seems contradictory and unsatisfactory.

---

## Referee Report (RR2)

**June 6th, 2025**

Insights into the North Hemisphere daily snowpack at high resolution from the new
Crocus-ERA5 product
By: S. Ramos Buarque et al.

**General Comments**

The authors have successfully addressed my previous comments, leading to a substantial improvement in the manuscript's quality.

However, there are several sections, particularly those providing background information, that lack sufficient citations.

I will highlight specific portions of the text that need attention below, but I recommend the authors thoroughly review the manuscript to ensure all statements are properly cited.

**Specific Comments**

1. L24–54: Please add the necessary citations, as this section currently contains none.

2. L83–105: This section also lacks citations; please address this in the same manner as point #1.

3. L192: State and define which statistics will be calculated (e.g., R and RMSE).

4. L449: Since the acronym SWE has already been defined, remove the redundant definition here and simply emphasize its importance to Arctic amplification.

---

## Referee Report (RR3)

The authors have thoughtfully addressed most of my primary concerns. From my perspective, the manuscript will be suitable for publication after the below two minor, but important, comments are addressed.

**Follow-up on Comment 1.4**: I appreciate the authors' response focused on the SCF/snow depth comparisons; however, it seems lacking. First, does this explanation (that relies on derivation of SCF/snow depth relationships) suggest that the model used in this study does not explicitly calculate SCF? If that is the case, how are SCF biases (e.g., in Figure 9) calculated? If the model does directly derive SCF, then the scheme used to calculate SCF in the simulations should be reported. Is the snow cover area shown in these plots from the model output; if so then there is clearly an SCF scheme used by the model, and that should be noted in the paper. These figures show distinct regimes for snow depth/SCF relationships for both shallow and deep snowpacks, likely from differences in SCF/snow depth relationships during accumulation vs. ablation periods, which is often parameterized in models based on snow density (e.g., Niu and Yang, 2007).

Niu, G.-Y., Yang, Z.-L., 2007. An observation-based formulation of snow cover fraction and its evaluation over large North American river basins. J. Geophys. Res. 112, D21101. https://doi.org/10.1029/2007JD008674

**Follow-up on Comment 1.7**: Although large scale temperature and precipitation forcing may govern interannual variability of SWE anomalies for a given area, land cover characteristics, such as land cover type, can drive large spatial heterogeneity. For example, the same meteorological conditions that occur in a forest could result in substantially different snowpack dynamics relative to a nearby grassy meadow. This limitation is important to address.

---

## Author Response (AR2)

**#essd-2024-451 - Author's response**

**Insights into the North Hemisphere daily snowpack at high resolution from the new Crocus-ERA5 product**

**General statement**

We are grateful to Reviewer #1 for the constructive comments and suggestions, and to Reviewer #2 for the supportive and focused feedback. We also thank the editor for overseeing the review process. Together, these contributions have helped us improve both the clarity and overall quality of the manuscript. Below, we provide detailed point-by-point responses, with our changes highlighted in blue for clarity.
* * *
**Reviewer 1**

**General Comment** — As noted in the original review, this paper can provide an important contribution; however, many of the key concerns expressed in the first round of review remain. Importantly concerns regarding how to meaningfully visualize results remain and there is a substantial lack of supporting citations throughout. Specific comments are included below.

**Reply**: Thank you for recognizing the value of our contribution. We addressed each of the specific instances you highlighted in the detailed responses below. Regarding the visualization concerns raised in your comments, we carefully revised each figure to align with your feedback and ensure clarity.

As for the references, we thoroughly reviewed the manuscript and expanded the citations to better situate our work within the existing literature.

**Reviewer Comment 1.1** — Introductory paragraphs require supporting citations (e.g., P2 and P3 have minimal citations).

**Reply**: Thank you for pointing this out. We have revised Paragraphs 2 and 3 to include several key references supporting the statements on sea ice, continental snow cover, permafrost, and Arctic feedback processes, as well as on Arctic monitoring efforts.

**Reviewer Comment 1.2** — Section 2, starting at L75: arguments made in this section should be strengthened with supporting citations. This sentiment should be considered throughout the article.

**Reply**: We have addressed this comment by adding supporting citations throughout Section 2 (starting at line 75) and across the rest of the manuscript to substantiate each key statement, ensuring that the arguments are now systematically backed by relevant and up-to-date references.

**Reviewer Comment 1.3** — L133: Since the early 2010s not "2010"

**Reply**: Done.

**Reviewer Comment 1.4** — Section 2.3: More detail on the calculation of evaluated variables is valuable (e.g., mathematical representation of a snow depletion curve used to calculate snow cover extent).

**Reply**: We appreciate the reviewer's suggestion. In response, we have expanded the description in Section 2.3 to clarify how snow cover extent is evaluated in our analysis. Rather than applying a predefined analytical snow depletion curve, we examined the modeled relationship between snow-covered fraction (SCF) and snow depth using spatially weighted outputs from both study domains.

The resulting scatterplot reveals two distinct linear behaviors, suggesting a dual-regime structure in the snow depletion process: one characterized by shallow snow depths with large SCF variability, and another where snow depth increases more rapidly with SCF.

This clear dual-linear structure is present in both regions analyzed — Eurasia and North America (Canada) — and is interpreted as evidence of two distinct snowpack regimes: (i) a shallow-snow regime, where SCF varies primarily due to patchy, residual snow cover; and (ii) a deeper-snow regime, where snow depth increases more rapidly with SCF — that is, the slope of the snow depth–SCF relationship is steeper — indicating the development of a continuous snowpack.

In the shallow-snow regime, the flatter slope suggests greater SCF sensitivity to surface heterogeneity and local bare-ground exposure than to snow depth itself. In contrast, the steeper slope in the deeper-snow regime reflects the influence of a more uniform, depth-driven snow cover.

The coexistence of these two regimes points to a transitional behaviour in snow depletion dynamics, marking a shift from patchy, remnant snow to extensive, continuous snowpack. This analysis highlights that the fraction of bare ground — rather than snow depth alone — is the main driver of SCF reduction during the melt period.

Although no additional figure has been included, this interpretation is now explicitly discussed in the revised text (Section 2.3) to clarify the physical basis of the snow depth-SCF relationship in the model outputs.

We believe this addition provides a physically grounded response to the reviewer's request for detail on how snow cover extent is evaluated in our study.

[Figure]

**Reviewer Comment 1.5** — Section 2.5: There is no guarantee that the in-situ snow depth observations, particularly with these observations tending to be in clearings, provide spatial representation allowing a fair comparison against simulation snow depth at a course (0.25-

degree) resolution. There needs to be a qualification of this point. The elevation difference was considered, which is good, but other heterogeneities (e.g., from land cover, aspect, wind redistribution, etc) can play important roles.

**Reply**: We agree with the reviewer that the in-situ measurements, while mostly taken over open ground in accordance with WMO standards, may not fully capture the variability within a 0.25° grid cell. To clarify this point, we have added the following sentence at the end of the relevant paragraph in Section 2.5: "While such site selection ensures good consistency with the open-field herbaceous configuration of Crocus-ERA5, it does not guarantee full spatial representativeness of a 0.25° grid cell. Sub-grid heterogeneities such as partial forest cover, slope, aspect, or wind-driven redistribution may still locally influence the comparison between observations and simulations."

**Reviewer Comment 1.6** — IMS has known uncertainties and this should be addressed.

**Reply**: We agree with the reviewer and have added a sentence in Section 2.5 acknowledging the known limitations of IMS, including cloud-snow discrimination issues, coarse resolution at high latitudes, and potential misclassification during transition seasons, while also justifying its use in this study.

**Reviewer Comment 1.7** — L230: The claim that forests do not have an important impact on interannual variability needs to be supported, proven, or removed. If removed, this limitation needs to be addressed.

**Reply**: We have rephrased the text to clarify that the conclusion is supported by the multi-product evaluation of Mudryk et al. (2025), which assessed 23 gridded SWE products, including Crocus-ERA5. Their results show that, for products without snow data assimilation and forced by the same meteorological reanalysis, interannual variability is primarily controlled by large-scale temperature and precipitation forcing, rather than by specific land cover representations. In this context, the absence of explicit snow–forest interactions in Crocus-ERA5 mainly affects the mean snow depth, while year-to-year anomalies and trends remain consistent with other top-performing products.

**Reviewer Comment 1.8** — L244: "Negative" rather than "Negatives"

**Reply**: Done.

**Reviewer Comment 1.9** — Figure 1: In my original comment I had requested anomaly diference plots, rather than only a presentation of the anomalies. This point remains important to visualize the diRerences between ERA5 and ERAI. Showing the anomalies, as currently done, is valuable for the reason the authors' noted in their original response, but visualizing diRerences is also very important to understand discrepancies between the 2 versions. I disagree with the authors' argument on maintaining the calendar year ordering, as noted in their response, the snow accumulation starts in Oct-Nov (i.e., the start of the water year), so it would make sense to present results in this manner. This note applies also to Figure 7.

**Reply**: As suggested, we have revised Figure 1 (and Figure 7) to now include plots of the monthly anomaly differences. This representation directly addresses the need to visualize systematic differences between the two reanalyses beyond the individual anomaly patterns. In the new Figure 1 — which shows ERA5 anomalies minus ERAI anomalies — the differences span a much narrower range (−0.02 m to +0.02 m) compared to the original anomalies (−0.1 m to +0.1 m). This required adjusting the color scale to preserve visual contrast and readability.

Regarding the time axis of the climatology, we acknowledge the reviewer's point concerning the snow accumulation season. To maintain consistency with other datasets and figures organized around the calendar year (e.g. Mortimer et al., 2020), we have initially retained the January–December ordering. However, we also recognize that aligning with the hydrological year (October–September) is more appropriate for interpreting snow-related processes, and we have therefore adopted this time structure in the revised figures.

**Reviewer Comment 1.10** — More clarification is needed for the presented anomaly calculation. For example, in panel 3 in 1995 there are negative snow anomalies in typical high SWE months (April-June) whereas SWE anomalies are positive in summer months when SWE tends to be minimal (Jul-Sept). In the calculation of the anomalies, are the respective monthly averages subtracted or the total time series average? If the latter, this result would suggest that SWE is relatively higher in summer months which would be perplexing. Thus, I assume the former is the case here which should be articulated.

**Reply**: We confirm that the monthly anomalies are computed by subtracting the *respective monthly climatologies* (1979–2018) from each dataset. That is, each month is compared to its own long-term monthly mean (e.g., April 1995 is compared to the April climatology), ensuring that the anomalies reflect deviations from typical seasonal behavior. This has been clarified in the revised manuscript. Regarding the example mentioned for 1995 in Panel 3, where *negative anomalies appear in April–June and positive anomalies in July–September*, this pattern is consistent with the method described above and can be explained as follows:

1. The **April–June negative anomalies** indicate that SWE in those months was *lower than the average SWE for those same months over 1979–2018*. This could result from below-average snowfall or earlier-than-usual melting during that year.

2. In contrast, **positive anomalies in July–September**, despite SWE typically being minimal, suggest that a small amount of residual snow remained in certain areas, *more than what is climatologically expected for summer*. In summer, because the climatological snow depth is close to zero, even minimal residual snow in the model may appear as a positive anomaly.

3. These summer anomalies are mainly driven by the model's internal snow processes (e.g., energy balance, compaction, metamorphism), which govern the persistence of snow in isolated regions. In such low-snow periods, small differences in model physics can generate noticeable anomaly signals, though their *absolute magnitudes remain low*.

We have revised the manuscript to clarify both the anomaly calculation method and this type of seasonal interpretation.

**Reviewer Comment 1.11** — Figure 2: Snow cover area, which is often represented as snow cover fraction in comparisons, is physically meaningful (e.g., for surface albedo) and allows comparisons across regions of diferent sizes. I maintain that showing the snow cover fraction here would be meaningful and would be a valuable supplementary figure.

**Reply**: I agree with the reviewer's point about the physical relevance of snow cover fraction (SCF) and its usefulness for interregional comparisons. However, while SCF is a meaningful variable at the grid-cell scale (ranging from 0 to 1), its use for hemispheric-scale anomaly time series is less straightforward. SCF is a fractional quantity defined at each model grid point and is not directly comparable across models or observations unless weighted by grid-cell area. It is difficult to aggregate or interpret SCF meaningfully over time without accounting for differences in grid cell area. By contrast, snow cover extent (SCE) provides an area-integrated metric that aligns more closely with the methodologies used in prior assessments and enables consistent comparisons of interannual and regional variability. Since SCE is derived from SCF, we have therefore retained the SCE-based approach in the main manuscript.

**Reviewer Comment 1.12** — Figure 2: ... The caption notes "Filled" circles, but the circles are empty.

**Reply**: It was a mistake. The caption has been corrected.

**Reviewer Comment 1.13** — Figure 2: ... In the original response, the authors' note that the figure format is selected to maintain consistency with prior studies to allow easy comparisons, which makes sense, however these 2 studies do not seem to be cited in this article.

**Reply**: The diagnostic we used – the Arctic SCE anomaly index – is consistent with the methodology applied in the NOAA *Arctic Report Card 2024* (available at: `https://arctic.noaa.gov/wp-content/uploads/2024/12/ArcticReportCard_full_report2024.pdf`), which was itself inspired by the approach of Callaghan et al. (2011, *Ambio*, `https://link.springer.com/article/10.1007/s13280-011-0212-y`). These studies analyze anomalies in snow cover extent (SCE) and duration (SCD). In the revised manuscript, we have explicitly cited both references to clarify the provenance of this diagnostic. Specifically, we compute standardized monthly SCE anomalies (relative to the 1991-2020 climatology) for the North American and Eurasian sectors ($\geq 60°$N) and apply 5-year running means to highlight long-term seasonal variability.

**Reviewer Comment 1.14** — Figure 4: I maintain that showing a difference plot is highly valuable here. If the authors worry that this would add too much complexity to this figure, then the difference can be shown in supplementary. The difference would be highly insightful to present observed vs. modeled differences in snow depth between climatological periods.

There is also a key issue with the current representation in Figure 4. Namely, it seems the goal with this figure is to show whether the model captures the spatial distribution of snow depth relative to observations, but in areas with overlapping circles that are filled with color, there is no way to see the underlying observation color in the map. Therefore, it would be appropriate to present a bias map of the circles (small enough to reduce overlap) colored by the model bias (e.g., using a polar color scheme from red to blue to present underestimates to overestimates).

**Reply**: We thank the reviewer for these suggestions. Indeed, the visual overlap between the filled observation circles and the underlying model field is a deliberate feature, designed to reveal agreement or disagreement between observed and modelled snow depth through colour correspondence. Observation circles are filled using the same color scale as the background model field: when observations and model values match, the circle blends with the background, while contrasting colors highlight discrepancies. For example, when the model overestimates snow depth relative to the observation (often in mountainous regions), the circle visually interrupts the field, drawing attention to this difference. To improve clarity, we have also added black outlines to all observation circles.

Regarding the suggestion to include a difference plot, we agree that it provides valuable insight. Thus, we have included a plot in the figure showing modeled minus observed climatological snow depth differences. The color scheme is designed to clearly highlight under- and overestimates, allowing readers to better assess spatial biases.

**Reviewer Comment 1.15** — Figure 7: I largely disagree with the authors' argument to not show SCF, which I consider potentially more informative than a comparison of anomalies due to the physical meaningfulness of SCF. Furthermore, an anomaly comparison will mask systematic biases which are important to present. If the authors would like to present standardized anomalies, this is reasonable, but the physically meaningful comparison of SCF, which has the ability to provide a presentation of systematic biases, should be included as well (e.g., in supplementary). The SCE comparison is useful but has lower granularity.

The original suggestion of showing the difference panel remains, as this is needed to easily visualize the discrepancies between the model and observations.

**Reply**: We reiterate our thanks to the reviewer for highlighting the importance of presenting the SCF, which provides a physically meaningful basis for identifying systematic biases. The manuscript already includes maps of SCF differences (Crocus-ERA5 minus IMS estimates, Figure 9), fulfilling the original suggestion to visualize discrepancies between model and observations. These maps show the seasonal mean SCF bias over the Northern Hemisphere and are now explicitly referred to as SCF difference maps (bias) in the revised captions and main text.

In response to the reviewer's comment, we have further clarified the relationship between SCF and SCE. While SCF is highly relevant at the grid-cell scale (ranging from 0 to 1), its aggregation into hemispheric-scale anomaly time series can be prone to errors, because SCF is defined per

grid cell and is not directly comparable across models or observations without accounting for grid-cell area. By contrast, SCE provides an area-integrated metric that allows robust comparisons of interannual and regional variability. Importantly, SCE is derived from SCF, and the larger aggregated values are statistically more robust.

Accordingly, the revised manuscript retains anomaly comparisons based on SCE for the monthly mean over the Arctic Northern Hemisphere, while also providing SCF difference maps (bias) to explicitly show physically meaningful deviations, as suggested by the reviewer. This ensures that both the granularity of SCF and the robustness of SCE are clearly communicated.

**Reviewer Comment 1.16** — Figure 9: The color bar label should note snow cover biases, rather than snow cover. A polarized color bar here is appropriate.

**Reply**: We have revised the figure accordingly. Initially, we used a categorical color scale to highlight low SCF values, because of it makes small differences more visible. However, following the reviewer's recommendation, we have revised Figure 9 to use a diverging (polarized) color scheme centered on zero, with the color bar label now explicitly indicating snow cover bias. This improves consistency and makes overestimations as underestimations easier to interpret.

**Reviewer Comment 1.17** — In the original response the authors' note that a detailed assessment for key snow-related variables in this data set (e.g., SWE and snow albedo) are beyond the scope of this study. Yet the opening sentence in the abstract is: "This article provides a detailed analysis of the Crocus-ERA5 snow product covering the Northern Hemisphere from 1950 to 2022". This seems contradictory and unsatisfactory.

**Reply**: We thank the reviewer for highlighting this ambiguity. The first sentence of the abstract has been revised to clearly state that the evaluation in this study focuses on snow depth and snow cover, rather than all variables available in the Crocus-ERA5 dataset. The revised sentence now reads: "This article provides an overview of the daily Crocus-ERA5 snow product covering the Northern Hemisphere from 1950 to 2022. It assesses the product's performance in terms of snow depth and cover compared to in situ observations and satellite data." This wording, we believe, resolves the perceived contradiction and aligns the abstract with the stated scope of the study.
* * *
**Reviewer 2**

**General Comment** — The authors have successfully addressed my previous comments, leading to a substantial improvement in the manuscript's quality.

However, there are several sections, particularly those providing background information, that lack sufficient citations.

I will highlight specific portions of the text that need attention below, but I recommend the authors thoroughly review the manuscript to ensure all statements are properly cited.

**Reply**: Thank you for recognizing the improvements made to the manuscript. In response to your comments, we thoroughly reviewed the background and contextual sections to ensure that all relevant statements are now supported by appropriate citations.

We also carefully revised the figures to enhance clarity and align with the feedback provided by Reviewer #1.

**Reviewer Comment 2.1** — L24–54: Please add the necessary citations, as this section currently contains none.

**Reply**: We appreciate the reviewer pointing out the lack of references in this section. We have now revised the text to include appropriate citations, adding both recent studies and key references that provide specific evidence and general background knowledge. These updates ensure that the discussion is properly supported by the relevant literature.

**Reviewer Comment 2.2** — L83–105: This section also lacks citations; please address this in the same manner as point #1.

**Reply**: We have revised the text accordingly. As in the Introduction, this section has been updated to include appropriate citations, drawing on both recent studies and foundational references to support the discussion.

In addition, a paragraph concerning SWE has been added in response to Reviewer #1 and has likewise been complemented with the relevant references.

**Reviewer Comment 2.3** — L192: State and define which statistics will be calculated (e.g., R and RMSE).

**Reply**: We have clarified this point in the revised manuscript. The following statistics are now explicitly stated and defined: correlation coefficient (R, strength of the linear relationship model-obs), bias (model minus observations), centered Root Mean Square Error (CRMSE, deviations after removing the mean bias) and, sample size (N, number of paired values used in the comparison).

**Reviewer Comment 2.4** — L449: Since the acronym SWE has already been defined, remove the redundant definition here and simply emphasize its importance to Arctic amplification.

**Reply**: The redundant definition of SWE has been removed, and the text now simply emphasizes its importance for Arctic amplification.

**Editor's comments**

Whilst one reviewer considers the manuscript to be greatly improved, the other maintains that several issues raised in their original review remain to be sufficiently addressed. These are largely related to the visualisation of the data.

In addition, both reviewers point out that references of existing literature in the introduction / background section require expanding.

I would therefore be grateful if you could revise the manuscript according to the feedback provided by the reviewers, focusing on these two aspects.

**Reply**: We thank the editor for summarizing the reviewers' feedback and for highlighting the key aspects requiring further revision. In response, we have carefully revised the manuscript with particular attention to:

- Data visualization – Figures have been updated and refined to improve clarity and accessibility, including the adoption of colorblind-friendly palettes, clearer labeling, and more explicit highlighting of under- and overestimates. These changes address the concerns raised by Reviewer 2 and enhance the overall readability of the results.

- Expanded references in the Introduction and background – We have substantially enriched the literature review, incorporating both recent studies and foundational works to strengthen the context and support of our findings.

We believe these revisions directly address the concerns raised and contribute to a clearer and more robust manuscript.

**Summary of manuscript changes**

- Added citations throughout the text, particularly in the Introduction and Section 2.

- Revised Figures 1 and 7 to illustrate differences between monthly anomalies.

- Updated Figure 2 for black-and-white readability.

- Modified Figure 4 to include a difference plot (bias).

- Revised Figure 9 to display snow cover fraction biases using a diverging scale.

- Adapted all figures to colorblind-friendly palettes for improved accessibility.

Overall, these revisions strengthen the consistency between the figures and the narrative, and improve both clarity and accessibility of the manuscript.

---

## Author Response (AR3)

**#essd-2024-451 - Author's response**

**Reviewer 1**

**General Comment** — The authors have thoughtfully addressed most of my primary concerns. From my perspective, the manuscript will be suitable for publication after the below two minor, but important, comments are addressed.

**Reply**: Thank you for recognizing the value of our contribution.

**Reviewer Comment 1.1** — Follow-up on Comment 1.4: I appreciate the authors' response focused on the SCF/snow depth comparisons; however, it seems lacking. First, does this explanation (that relies on derivation of SCF/snow depth relationships) suggest that the model used in this study does not explicitly calculate SCF? If that is the case, how are SCF biases (e.g., in Figure 9) calculated?
If the model does directly derive SCF, then the scheme used to calculate SCF in the simulations should be reported. Is the snow cover area shown in these plots from the model output; if so then there is clearly an SCF scheme used by the model, and that should be noted in the paper. These figures show distinct regimes for snow depth/SCF relationships for both shallow and deep snowpacks, likely from diIerences in SCF/snow depth relationships during accumulation vs. ablation periods, which is often parameterized in models based on snow density (e.g., Niu and Yang, 2007).

Niu, G.-Y., Yang, Z.-L., 2007. An observation-based formulation of snow cover fraction and its evaluation over large North American river basins. J. Geophys. Res. 112, D21101. https://doi.org/10.1029/2007JD008674

**Reply**: We thank the reviewer for the opportunity to clarify how SCF is handled in Crocus-ERA5 and how to interpret the SCF–snow-depth relationships shown.
In the Crocus-ERA5 product, the model is run in an "open-field" configuration (Brun et al., 2013): snow–vegetation interactions are disabled and forests are not represented. In this setting, the snow-covered fraction (SCF) is not parameterized at sub-grid scale; instead, it is diagnosed in a simple, quasi-binary manner from snow water equivalent (SWE):

$$SCF = min(1, \frac{W_{sn}}{W_{lim}})$$

where $W_{sn}$ is the snow mass (kg m$^{-2}$) and $W_{lim}$ set to 1 kg m$^{-2}$. Hence, as soon as a thin snowpack is present, SCF rapidly approaches 1. We adopt this choice since the previous Crocus-ERAI product of Brun et al. (2013) for the reasons set out below.

(i) Our aim is to resolve intrinsic snowpack properties (stratigraphy, metamorphism, albedo, thermal properties, etc.), which would be diluted by a surface-averaged mixture including snow-free areas; a quasi-binary SCF avoids that dilution. This logic is also consistent with satellite products such as IMS, whose SCF is binary at its grid resolution (1/4/24 km: 1 if snow is present, 0 otherwise).

(ii) Consequently, comparisons with IMS are like-for-like: our SCF diagnostic targets presence/absence of snow at the grid scale, as IMS does.

(iii) Finally, following Brun et al. (2013) in the Crocus-ERAI replay, the "open-field" setup facilitates comparison with local in-situ observations in open terrain and provides a more direct physical meaning, avoiding uncertainties from snow–vegetation interactions that are not represented in our framework.

There was therefore an error in the previous version of the manuscript, on page 6 (lines 174-178). The text has now been corrected and improved (lines 168-183). The main changes are as follows:

"Because these interactions are complex and remain challenging to simulate accurately, Crocus-ERA5 adopts a simplified "open-field" configuration, simulating the snowpack over an idealised grassland cover with climatological physiography (e.g., no interannual variability in leaf area index). In this configuration, the SCF is computed directly from the simulated SWE rather than from a vegetation, depth, or density dependent sub-grid scheme. Crocus-ERA5 applies a ramp-to-one diagnostic: SCF increases linearly with snow mass up to a small threshold (1 kg m$^{-2}$ of SWE) and saturates at 1 beyond it. This design centers the evaluation on intrinsic snowpack properties and ensures consistency with both the binary IMS product at its grid resolution and local open-field in-situ observations. For consistency, the IMS product was processed to obtain a comparable continuous SCF. Daily categorical values (water, land, ice, and snow) were averaged over time, producing intermediate values that reflect the frequency of snow occurrence within each grid cell. These mean values were linearly rescaled to express SCF as a continuous percentage. Despite their distinct formulations - physical for Crocus-ERA5 and statistical for IMS - both approaches provide consistent measures of snow cover extent suitable for climatological comparison. The snow cover fraction (SCF) was derived consistently from both model and observational datasets, each according to its native definition. These mean values were linearly rescaled to express SCF as a continuous percentage. Despite these conceptual differences, both estimates provide compatible measures of snow cover extent suitable for climatological comparison."

Instead of:

"Crocus-ERA5 addresses this by simulating snowpack evolution in detail, including the snow cover fraction (SCF), and by distinguishing snow-covered from snow-free areas (e.g., vegetation or bare soil) within each grid cell (Brun et al., 2013). This separation limits unrealistic vegetation–snow interactions and improves the simulation of snowmelt and rain infiltration."

Consequently, SCF biases calculation (e.g., in Figure 9) are computed as the difference between the model-diagnosed SCF and the IMS-derived SCF at the grid scale. Theses biases mainly stem from: **(i)** ERA5 forcing uncertainties (precipitation amount/phase, temperature), **(ii)** timing of melt in Crocus (too early or too late), and **(iii)** representativeness differences between our quasi-binary SCF and the binary IMS product.

On the dual-regime structure (SCF–snow depth), the observed two-regime pattern for shallow vs. deep snow does not arise from an active sub-grid SCF scheme in our configuration. It reflects the quasi-binary nature of our SCF diagnostic combined with the temporal evolution of SWE through accumulation and ablation. This is consistent with hysteresis-like behaviors reported in the literature (e.g., Niu and Yang, 2007), even though we do not parameterize SCF as a function of snow density or vegetation here.

**Reviewer Comment 1.2** — Follow-up on Comment 1.7: Although large scale temperature and precipitation forcing may govern interannual variability of SWE anomalies for a given area, land cover characteristics, such as land cover type, can drive large spatial heterogeneity. For example, the same meteorological conditions that occur in a forest could result in substantially different snowpack dynamics relative to a nearby grassy meadow. This limitation is important to address.

**Reply**: We thank the reviewer for the clarification. We would like to emphasize a point about scale: the original Comment 1.7 addressed interannual variability (i.e., year-to-year anomalies), whereas the follow-up comment focuses on spatial heterogeneity driven by land cover. These are related but distinct issues.

Our initial statement remains valid in the stated context: for non-assimilative products forced by the same reanalysis, the interannual SWE anomaly signal is primarily controlled by large-scale forcing (temperature and precipitation). The lack of an explicit snow–forest scheme mainly affects mean levels (SWE/HS biases) rather than the temporal coherence of anomalies and trends at large scales. Conversely, at local scales, we agree that land-cover type (forest vs. grassland) produces strong spatial contrasts in snow processes (interception, shading, wind, sublimation, etc.) that our "open-field" setup does not represent—this is indeed a limitation that we state explicitly.

We also note that land-cover changes (e.g., logging, conversion to cropland) can indeed affect interannual variability at the time they occur. However, with constant land cover, spatial heterogeneity should not be conflated with interannual variability: the former concerns differences across sites, the latter evolution over time under the same forcing.

We will revise the end of the conclusion of the manuscript to:

 (i) make this temporal vs. spatial distinction explicit;

 (ii) bound the scope of our conclusions (robust for large-scale anomalies over open terrain); and

(iii) clearly state that local forest–meadow contrasts are not reproduced in our configuration and that averages over heavily forested areas may be biased.

We will add the following paragraph (last version, line 554):

"In this context, it is important to clearly delineate the limitations of the Crocus-ERA5 product, particularly those arising from the lack of explicit snow–forest interactions. Building on the comparison above, we emphasize the distinction between temporal and spatial aspects of performance. Our conclusions primarily concern interannual anomalies driven by large-scale temperature and precipitation forcing in an open-field configuration (non-forested, grass-dominated surfaces), for which Crocus-ERA5 reproduces year-to-year variability robustly across most regions. By contrast, local spatial contrasts associated with land cover, such as differences between forest and meadow, are not represented in our setup (canopy interception, shading, wind sheltering, and enhanced sublimation are absent). Consequently, forest-meadow differences are not reproduced, and absolute values of SWE and snow depth over heavily forested areas may be biased. These limitations should be considered when interpreting spatial patterns, while our findings on large-scale anomalies over open terrain remain robust. Notably, Mudryk et al. (2025) recently showed that Crocus-ERA5 ranks among the most effective product for reproducing SWE in the Northern Hemisphere, at least in plain areas."

---

## Author Response (AR4)

**Response to Reviewers**

Manuscript ID: **ESSD-2024-451**
*Insights into the North Hemisphere daily snowpack at high resolution from the new Crocus-ERA5 product*

Dear Dr. Thornton,

Thank you for forwarding the reviewer's further remarks and for managing the review process. We are grateful for the reviewer's careful re-reading.

We have carefully addressed all remaining comments, revised the text accordingly, and updated figures where needed. Our detailed, point-by-point responses to each reviewer's comments are provided below.

We would also like to thank the reviewer for their thoughtful and constructive feedback, which has helped improve the clarity and quality of the manuscript.

Sincerely,

Silvana R. Buarque,
Bertrand Decharme,
Alina L. Barbu,
Laurent Franchisteguy

**Summary of Revisions**

The structure of the manuscript has changed considerably during the revision process. Several paragraphs have been moved to Section 2 (Data and Methods) to improve clarity and organization. Because these modifications involved large text block relocations rather than isolated edits, a point-by-point explanation of each movement would not be informative. For this reason, we provide below a structured summary of the main revisions rather than detailed explanations of each textual change.

- **Abstract & Introduction**

  - Clarified model limitations in low-vegetation areas (e.g., tundra) and potential biases.

  - Rewrote part of the introduction to highlight the novelty of the Crocus-ERA5 product.

  - Improved justification for using snow depth instead of SWE, with added methodological explanation.

- **Methods Section**

  - Expanded description of observational datasets (uncertainties, scale mismatch).

  - Clarified methodology for SCF calculations.

  - Added discussion of model limitations in boreal forest.

  - Enhanced description of statistical methods and evaluation criteria.

- **Figures & Data Presentation**

  - Revised Figures 1 and 7 to include anomaly difference plots and adopt the water year for climatology.

  - Improved clarity of Figures 2 and 4 (labels, legends, standardized anomalies, added difference panel for biases).

  - Updated Figures 9: diverging color scales for biases.

- **Text Revisions**

  - Revised or removed statements about boreal forest applicability.

  - Expanded discussion on snow ablation (melting + sublimation).

  - Clarified anomaly calculation and standardization rationale.

  - Added several references throughout Introduction and Methods.

- **SCF Handling**

- Corrected description of SCF calculation: Crocus-ERA5 uses quasi-binary SCF derived from SWE.
- Updated text to explain SCF bias computation and rationale.

- **Statistical & Uncertainty Treatment**

  - Clarified p-value interpretation.
  - Added details on MAE calculation for standardized vs non-standardized anomalies.
  - Acknowledged uncertainties in IMS satellite data and in-situ representativeness.

- **References & Citations**

  - Added missing references in key sections.
  - Explicitly cited NOAA Arctic Report Card 2024 and Callaghan et al. (2011).

- **Conclusion**

  - Clarified distinction between temporal (interannual) and spatial (land cover) variability impacts.
  - Emphasized scope: Crocus-ERA5 is robust for open-field conditions, not for forested areas.
* * *
**Total of revisions involved**

- Abstract + 3 major sections revised.

- Figures: 1, 2, 4, 7, 9 updated.

- Clarifications on SCF calculation and biases.

- Improved methodological transparency and references.

- Strengthened text regarding limitations and applicability